# Advances in representing interactive methane in ModelE2-YIBs (version 1.1)

Kandice L. Harper[1], Yiqi Zheng[2], Nadine Unger[3]

[1]School of Forestry and Environmental Studies, Yale University, New Haven, CT, 06511, USA
[2]Department of Geology and Geophysics, Yale University, New Haven, CT, 06511, USA
[3]College of Engineering, Mathematics and Physical Sciences, University of Exeter, Exeter, EX4 4QJ, UK

*Correspondence to*: Kandice L. Harper (kandice.harper@yale.edu)

**Abstract.** Methane ($CH_4$) is both a greenhouse gas and a precursor of tropospheric ozone, making it an important focus of chemistry–climate interactions. Methane has both anthropogenic and natural emission sources, and reaction with the atmosphere's principal oxidizing agent, the hydroxyl radical (OH), is the dominant tropospheric loss process of methane. The tight coupling between methane and OH abundances drives indirect linkages between methane and other short-lived air pollutants and prompts the use of interactive methane chemistry in global chemistry–climate modeling. In this study, an updated contemporary inventory of natural methane emissions and the soil sink is developed using an optimization procedure that applies published emissions data to the NASA GISS ModelE2-Yale Interactive terrestrial Biosphere (ModelE2-YIBs) global chemistry–climate model. Methane observations from the global surface air-sampling network of the Earth System Research Laboratory (ESRL) of the U.S. National Oceanic and Atmospheric Administration (NOAA) are used to guide refinement of the natural methane inventory. The wetland methane flux is calculated as a best fit; thus, the accuracy of this derived flux assumes accurate simulation of methane chemical loss in the atmosphere and accurate prescription of the other methane fluxes (anthropogenic and natural). The optimization process indicates global annual wetland methane emissions of 140 Tg $CH_4$ $y^{-1}$. The updated inventory includes total global annual methane emissions from natural sources of 181 Tg $CH_4$ $y^{-1}$ and a global annual methane soil sink of 60 Tg $CH_4$ $y^{-1}$. An interactive-methane simulation is run using ModelE2-YIBs, applying dynamic methane emissions and the updated natural methane emissions inventory that results from the optimization process. The simulated methane chemical lifetime of 10.4 ± 0.1 years corresponds well to observed lifetimes. The simulated year 2005 global-mean surface methane concentration is 1.1 % higher than the observed value from the NOAA ESRL measurements. Comparison of the simulated atmospheric methane distribution with the NOAA ESRL surface observations at 50 measurement locations finds that the simulated annual methane mixing ratio is within 1 % (i.e., +1 % to -1 %) of the observed value at 76 % of locations. Considering the 50 stations, the mean relative difference between the simulated and observed annual methane mixing ratio is a model overestimate of only 0.5 %. Comparison of simulated annual column-averaged methane concentrations with SCIAMACHY satellite retrievals provides an independent post-optimization evaluation of modeled methane. The comparison finds a slight model underestimate in 95 % of grid cells, suggesting that the applied methane source in the model is slightly underestimated or the model's methane sink strength is

slightly too strong outside of the surface layer. Overall, the strong agreement between simulated and observed methane lifetimes and concentrations indicates that the ModelE2-YIBs chemistry–climate model is able to capture the principal processes that control atmospheric methane.

# 1 Introduction

Atmospheric methane ($CH_4$) is a greenhouse gas that warms the climate by absorbing terrestrial radiation. The industrial-era increase in the methane concentration (+150 %) has induced a global-mean radiative forcing (+0.48 ± 0.05 W m$^{-2}$) that is the second largest in magnitude among all well-mixed greenhouse gases, smaller only than that induced by the increase in atmospheric carbon dioxide ($CO_2$, +1.82 ± 0.019 W m$^{-2}$) (Myhre et al., 2013). On a 20 year time scale, the global warming potential of methane is a factor of 84 larger than that for $CO_2$ (Myhre et al., 2013). In addition to its role as a climate forcer,
methane affects air quality through its role as a precursor of the harmful air pollutant tropospheric ozone (West and Fiore, 2005).

Methane is emitted to the atmosphere by both anthropogenic and natural sources (Ciais et al., 2013; EPA, 2010; Kirschke et al., 2013), including incomplete combustion of fossil fuels, biofuels, and plant biomass; seepage from terrestrial and marine
reservoirs; and through the action of methanogenic bacteria, which produce methane through anaerobic breakdown of organic matter. Methane generation through bacterial decomposition of organic matter occurs in: wetland soils; waterlogged agricultural soils, such as rice paddies; landfills; and in the digestive systems of ruminant animals and termites (Cicerone and Oremland, 1988). Removal of atmospheric methane occurs primarily through oxidation by the hydroxyl radical (OH), the atmosphere's principal oxidizing agent (Logan et al., 1981). Additional chemical loss occurs in the stratosphere via reaction
with chlorine radicals and excited-state oxygen radicals ($O^1D$) (Kirschke et al., 2013; Portmann et al., 2012). Uptake and oxidation of methane by methanotrophic bacteria in dry, aerated soils serves as an additional small sink (Kirschke et al., 2013).

The contemporary methane abundance and growth rate are well known owing to high-precision surface observations made
by global monitoring networks, such as that coordinated by the Earth System Research Laboratory/Global Monitoring Division (ESRL/GMD) of the National Oceanic and Atmospheric Administration (Dlugokencky et al., 2015). Methane chemical lifetime is not directly measured in the atmosphere, but has been derived from knowledge of the synthetic compound methyl chloroform ($CH_3CCl_3$; Prather et al., 2012; Prinn et al., 2005; Rigby et al., 2013). Methyl chloroform has well-known anthropogenic emissions and no natural emission source. Similar to methane, the principal sink of atmospheric
methyl chloroform is oxidation by OH. Observations of methyl chloroform abundance, in conjunction with estimates of methyl chloroform emissions, provide a means to estimate global OH abundance, methyl chloroform lifetime, and, subsequently, methane lifetime (Prinn et al., 1995). Together, these estimates provide a constraint on the total methane flux

into the atmosphere; however, apportionment of this total into contributions from the individual source sectors is highly uncertain (Kirschke et al., 2013; Saunois et al., 2016).

Because reaction with OH is the primary sink of methane, a change in the abundance of OH can alter methane's atmospheric burden and lifetime and, consequently, its capacities to both influence climate and generate ozone (Fry et al., 2012; Fuglestvedt et al., 1996). Emissions of nitrogen oxides ($NO_X$) decrease methane by increasing the oxidation capacity of the atmosphere, while emissions of non-methane volatile organic compounds (NMVOCs) and carbon monoxide (CO) increase methane by consuming atmospheric OH (Fry et al., 2012; Naik et al., 2005). Increased emissions of methane can prolong methane's own atmospheric lifetime (Fuglestvedt et al., 1996). Methane emissions can likewise influence the concentrations of other climate forcing pollutants; for example, the atmospheric burden of sulfate aerosols is influenced not only by emissions of the precursor gas sulfur dioxide ($SO_2$), but also by emissions of CO, $CH_4$, NMVOCs, and $NO_X$, which influence the conversion of $SO_2$ to sulfate aerosols by affecting the burdens of a variety of tropospheric oxidants (Shindell et al., 2009; Unger et al., 2006).

The strong oxidant-driven linkages among the short-lived air pollutants demonstrate the need to use global modeling to study chemistry–climate interactions, including those involving methane. In chemistry–climate model simulations, atmospheric methane is commonly represented through prescription of its surface concentration (Naik et al., 2013). Simulations using interactive methane (Shindell et al., 2013), in which the online methane concentration is dynamically tied to oxidant availability, can provide an improved understanding of chemistry–climate interactions. A spatially explicit methane emissions inventory is necessary for running interactive climate simulations that apply dynamic methane emissions. In this study, published sector-specific data on natural methane fluxes (Ciais et al., 2013; Dutaur and Verchot, 2007; EPA, 2010; Etiope et al., 2008; Fung et al., 1991; Kirschke et al., 2013; Melton et al., 2013; Saunois et al., 2016; Schwietzke et al., 2016) are used in conjunction with atmospheric modeling and atmospheric methane observations (Dlugokencky et al., 2015) to guide development of a spatially explicit contemporary budget of natural methane emissions and the methane soil sink. The NASA ModelE2-Yale Interactive terrestrial Biosphere (ModelE2-YIBs) global chemistry–climate model (Schmidt et al., 2014; Shindell et al., 2013; Yue and Unger, 2015) is subsequently used to run an interactive methane simulation representative of year 2005 that applies the refined natural methane flux inventory. The simulated atmospheric methane distribution is evaluated against multiple observational datasets. Because methane is an ozone precursor, a comparison of simulated ozone mixing ratios with a contemporary ozone climatology is also presented.

**2 Interactive methane in ModelE2-YIBs**

Atmospheric modeling, using ModelE2-YIBs, was used to develop an updated natural methane emissions inventory. The updated inventory is required for global chemistry–climate simulations that employ interactive methane emissions. A three-step methodology was applied. First, gridded input files of the natural methane emission sources and soil sink were built

using published inventories and flux information (Ciais et al., 2013; Dutaur and Verchot, 2007; EPA, 2010; Etiope et al., 2008; Fung et al., 1991; Kirschke et al., 2013; Melton et al., 2013; Saunois et al., 2016; Schwietzke et al., 2016). Secondly, ModelE2-YIBs simulations were performed; the simulations applied the natural methane emissions inventory and year 2005 emissions for all other emission sources of short-lived air pollutants. ModelE2-YIBs is described in Sect. 2.1, and the interactive methane simulation configuration and forcing datasets are described in Sect. 2.2. Thirdly, the modeled atmospheric methane distribution resulting from the second step was compared to methane surface observations at 50 globally distributed locations. The NOAA ESRL methane measurements (Dlugokencky et al., 2015) are described in Sect. 4. The model–measurement comparison was used to refine the spatial and temporal distribution of methane emissions from wetlands. The second and third steps were repeated, applying the newly optimized wetland emissions to ModelE2-YIBs, until strong model–measurement agreement was achieved. The resulting natural methane emissions inventory is described in Sect. 3, along with additional details about the optimization process for the wetland methane source. Evaluation of the simulated methane distribution based on the final inventory is presented in Sect. 4. Comparison of the modeled methane distribution with column-averaged methane concentrations derived from SCIAMACHY satellite retrievals (Schneising et al., 2009) serves as an independent validation of the simulated methane distribution.

Using ModelE2, Shindell et al. (2013) previously used a similar procedure of modifying the wetland methane source to achieve a modeled methane concentration that is in line with present-day observations, noting that the accuracy of the magnitude of the wetland flux that is derived in this way depends on whether the other prescribed fluxes have been accurately assigned. That is, the applied methodology calculates the wetland methane emission magnitude as a best fit under the assumption that the other methane fluxes and simulated atmospheric chemical loss are accurately represented in the global model. Relative to the Shindell et al. (2013) study, this study updates the natural non-wetland methane fluxes; applies a different anthropogenic emissions inventory; includes a new land surface model with interactive computation of isoprene and monoterpene emissions (Unger et al., 2013; Yue and Unger, 2015); and applies observed ocean boundary conditions. This methodology permits harmonization of the modeled methane mole fractions with contemporary observations, but can potentially misattribute the methane fluxes among the various source categories. Planned chemistry–climate simulations that will make use of the natural methane inventory developed here are specifically designed to investigate perturbations in anthropogenic methane emissions (i.e., the natural methane fluxes will be held constant using the magnitudes and distributions determined here). Any inaccuracies in assignment of the methane fluxes among the natural source sectors are relatively unimportant for the purposes of such studies.

The model input files prescribing the natural non-wetland methane sources have been developed based on the best available information (Sect. 3). For estimates of the global annual wetland methane flux, a recent model inter-comparison reported variation of ± 40 % around the multi-model mean for seven models that were driven with the same climate conditions and atmospheric $CO_2$ concentrations (Melton et al., 2013). It is because of the large uncertainty in the contemporary magnitude

of the wetland methane flux (Kirschke et al., 2013; Melton et al., 2013) that the emissions from this sector are optimized using atmospheric modeling.

## 2.1 Model description

The ModelE2-YIBs global chemistry–climate model is the result of the two-way coupling of the YIBs land surface model
(Yue and Unger, 2015) with the NASA GISS ModelE2 general circulation model (Schmidt et al., 2014). ModelE2-YIBs has a horizontal resolution of 2° latitude × 2.5° longitude with 40 vertical layers covering the global atmosphere from the surface to the 0.1 hPa model top. Physical and chemical processes are computed at a 30 minute time step.

The atmospheric chemical mechanism features 51 chemical species participating in 156 chemical reactions (Schmidt et al.,
2014; Shindell et al., 2006). Twenty seven chemical tracers are advected according to the model dynamics (Shindell et al., 2006). The troposphere and stratosphere are coupled in terms of both dynamics and chemistry (Shindell et al., 2006). Stratospheric chemistry includes nitrous oxide ($N_2O$) and halogen chemistry (Shindell et al., 2006). The troposphere includes standard $NO_X$-$O_X$-$HO_X$-CO-$CH_4$ chemistry; methane, isoprene, monoterpenes (as α-pinene), and formaldehyde are explicitly represented in the model, while other hydrocarbons are represented using a lumped scheme (Houweling et al., 1998) that is
based on the Carbon Bond Mechanism-4 (Gery et al., 1989) and the Regional Atmospheric Chemistry Model (Stockwell et al., 1997). More recent updates to the chemical mechanism are described by Shindell et al. (2006, 2013). The alkane and alkene lumped hydrocarbon classes used in the ModelE2-YIBs chemical mechanism are calculated from the total NMVOC emissions from the prescribed emissions scenario (described in Sect. 2.2) by applying spatially explicit alkane-to-total-NMVOC and alkene-to-total-NMVOC ratios from the RCP8.5 inventory (Riahi et al., 2011) for year 2005.

In this study, methane is calculated as an interactive tracer that is driven by methane surface fluxes, is influenced by oxidant chemistry, and, in turn, affects online oxidant availability (Shindell et al., 2013). This paper describes the new version 1.1 of ModelE2-YIBs. ModelE2-YIBs version 1.1 refers to the use of interactive methane chemistry and dynamic methane emissions (including application of the final contemporary natural methane flux inventory described in Sect. 3) within the
framework of ModelE2-YIBs version 1.0. ModelE2-YIBs version 1.0 refers to YIBs version 1.0 (Yue and Unger, 2015) coupled to the version of ModelE2 described by Schmidt et al. (2014). For anthropogenic and biomass burning sectors, emissions are prescribed for reactive gas and primary aerosol species. Biomass burning emissions are mixed into the atmospheric boundary layer. Vertically resolved $NO_X$ aviation emissions are injected at 25 levels that extend to an altitude of ~ 15 km. Prescribed emissions from all sectors other than biomass burning and aviation are treated as surface fluxes. Daily
surface fluxes are interactively interpolated from the relevant monthly or annual prescribed fluxes.

Climate-sensitive interactive emissions include: isoprene (Arneth et al., 2007; Unger et al., 2013), monoterpenes (Lathière et al., 2006), mineral dust (Miller et al., 2006), oceanic dimethyl sulfide (Koch et al., 2006), sea salt particles (Koch et al.,

2006), and lightning $NO_X$ (Price et al., 1997). Interactive radiatively active secondary inorganic aerosols include nitrate (Bauer et al., 2007) and sulfate (Koch et al., 2006). Secondary organic aerosols are formed from the interactive emissions of isoprene, monoterpenes, and other reactive volatile organic compounds (Tsigaridis and Kanakidou, 2007). Gas-phase aerosol precursors and oxidants affect the production and processing of aerosols (Bell et al., 2005), and aerosol-induced perturbations to the radiation budget impact photolysis rates (Bian et al., 2003). The online climate state provides the meteorological parameters that affect atmospheric chemistry, such as humidity, temperature, and sunlight. ModelE2 has previously undergone rigorous validation of simulated present-day tropospheric and stratospheric chemical composition and circulation (Shindell et al., 2006, 2013). Extensive evaluation of the atmospheric methane distribution that is simulated using the updated inventory of contemporary natural methane fluxes is presented in Sect. 4.

## 2.2 Simulation configuration

The atmosphere-only, time-slice simulation E2005 is representative of year 2005 and is run using interactive methane chemistry, including the use of dynamic methane emissions. The simulations were performed on the Omega cluster at the Yale Center for Research Computing (https://research.computing.yale.edu/support/hpc/clusters/omega). Omega is a 704-node 5632-core cluster based on the Intel Nehalem nodes and equipped with 36GB of RAM per node, a QDR Infiniband interconnect, and a high-speed Lustre DDN file system for parallel I/O. When the cluster was operating at peak performance, NASA ModelE2-YIBs had a runtime of 8–10 model days per hour using 88 processors.

Two datasets are used to define global anthropogenic and biomass burning emissions of the short-lived air pollutants for 2005: (1) a scenario derived from the Greenhouse gas–Air pollution Interactions and Synergies (GAINS) integrated assessment model (Amann et al., 2011; http://gains.iiasa.ac.at) and (2) the RCP8.5 emissions scenario (Riahi et al., 2011). GAINS emission scenarios are composed of three basic elements (Amann et al., 2011): (1) activity pathways that describe the temporal evolution of polluting activities; (2) region-specific emission factors for all emitted pollutants from all polluting activities; and (3) control strategies that define the degree of penetration of available pollution control technologies over time. The GAINS-derived global scenario for the short-lived air pollutants was created by combining existing scenario elements from the GAINS database: the activity pathway for the agriculture sector is based on estimates by the Food and Agriculture Organization (Alexandratos and Bruinsma, 2012) and those for the industrial process, mobile transport, and VOC-specific sectors are based on projections from the International Energy Agency (IEA, 2011); the energy sector activity pathway includes regional-level data from China (Zhao et al., 2013); and the pollution control strategy makes use of extensive updates for methane emission sources (Höglund-Isaksson, 2012).

The GAINS air pollution emissions scenario defines emissions from the anthropogenic sectors: agriculture, agricultural waste burning, domestic, energy, industrial, solvents, transportation, and waste. As the GAINS integrated assessment model does not project emissions from aviation, international shipping, or biomass burning (savanna and grassland fires and forest

fires) sectors, the E2005 simulation assigns the RCP8.5 emissions of short-lived climate pollutants and their precursors for these sectors (Riahi et al., 2011). Information from the GAINS model was used to develop the trajectory of future air pollution emissions in the RCP8.5 scenario (Riahi et al., 2011). Prescribed global annual-mean surface-level mixing ratios of the non-methane well-mixed greenhouse gases are likewise from the RCP8.5 scenario (Meinshausen et al., 2011; Riahi et al.,

2007): 379.3 ppmv $CO_2$, 319.4 ppbv $N_2O$, and 793 pptv chlorofluorocarbons (CFCs = CFC-11 + CFC-12).

Prescribed monthly-varying sea ice concentrations and sea surface temperatures are derived from the global observation-based Hadley Centre Sea Ice and Sea Surface Temperature dataset (Rayner et al., 2003), using averages over the years 2003–2007. The simulated concentrations of ozone, methane, and aerosols are allowed to affect the model radiation and, therefore,

meteorology and dynamics. In other words, these simulations allow rapid adjustments to the climate system (Myhre et al., 2013), and such climate perturbations can, in turn, affect the simulated atmospheric composition.

For simulations using the interactive methane scheme in ModelE2, the atmospheric methane distribution at initialization is defined through application of a vertical gradient, derived from HALOE observations (e.g., Russell et al., 1993), to

prescribed hemispheric-mean surface methane concentrations (Dlugokencky et al., 2015). The E2005 simulation applies the final contemporary natural methane flux inventory described in Sect. 3 that was developed using the optimization process. For most sectors, anthropogenic and natural methane emissions are prescribed in the climate model using global, gridded input files; lake, oceanic, and terrestrial geological methane emissions are internally calculated by the model through prescription of emission factors in the model source code. Using an interactive methane configuration with dynamic methane

emissions, the simulated atmospheric methane mixing ratio is temporally and spatially variable.

The E2005 simulation was run until atmospheric methane reached steady state, such that the global chemical sink came into balance with the net global source (prescribed sources minus prescribed soil sink), resulting in a relatively stable atmospheric methane abundance. Steady-state conditions were diagnosed using the global annual-mean atmospheric burden of methane.

The final 10 years of the 45 year simulation are used for analysis. Year-to-year variation in the methane burden for the final 10 model years is < 3.2 Tg $CH_4$. Year-to-year variation in the global-average surface methane concentration is < 1.3 ppbv. The year of interest for this study, 2005, fell within a roughly 8 year period that witnessed a largely stable global-mean concentration of methane in Earth's atmosphere (Dlugokencky et al., 2009). The observed stability in the concentration of methane does not necessarily indicate temporally invariant global sources and sinks over this era (Rigby et al., 2017; Turner

et al., 2017). For example, a recent analysis by Turner et al. (2017) suggests that simultaneous counterbalancing changes in methane emissions and loss to OH may be responsible for the observed stability in the methane concentration in the early 2000s. Therefore, the methane budget derived in this study by assuming steady state conditions represents just one plausible solution that can lead to a stable atmospheric methane concentration. This assumption is convenient in global chemistry–climate modeling where the simulated climate state does not correspond to an exact meteorological year. The derived

solution is constrained by both the prescribed methane fluxes and other forcing data that can affect atmospheric methane, such as: emissions of other short-lived compounds; the prescribed ocean conditions, which influence the physical climate state; and the concentrations of the non-methane long-lived greenhouse gases, which influence the radiation budget. The non-wetland natural methane fluxes that are prescribed are based on published estimates (Sect. 3) and are representative of the 2000s contemporary era but are not necessarily specific to year 2005. Likewise, the prescribed sea ice distribution and sea surface temperatures are observation-based five year means centered on year 2005. The derived methane budget, therefore, represents a 2000s climatology and is approximately, but not precisely, representative of year 2005 conditions.

The global annual emission magnitudes of the non-methane short-lived air pollutants for E2005 are summarized in Table 1; the methane budget is discussed in Sect. 3. The global annual-mean surface air temperature for E2005 is 14.6 ± 0.03 °C (average ± 1 standard deviation, calculated over 10 model years).

## 3 Contemporary natural methane emissions and soil sink

The contemporary natural methane budget used in this study is shown in Table 2. The non-wetland natural methane fluxes are derived from published estimates. The wetland methane emissions shown in Table 2 are the final result of the iterative optimization process introduced in Sect. 2 and described in more detail below.

Many of the natural methane emission input files used here were created by updating gridded emission files from a dataset produced by Fung et al. (1991). To construct best estimates of the spatial and temporal distribution of methane fluxes for the 1980s, Fung et al. (1991) first combined flux measurements, isotopic profiles, and land surface data to generate plausible flux scenarios and then refined the resultant scenarios using tracer transport modeling in conjunction with observations of the atmospheric methane concentration. For the natural methane budget in this project, the spatial distribution of the fluxes prescribed by Fung et al. (1991) was largely retained for most sources and for the soil sink, while the regional or global flux totals were scaled to match more recent estimates.

Global anthropogenic methane emissions for 2005 from the GAINS scenario are 325.1 Tg $y^{-1}$. This total excludes emissions from international shipping, which are not quantified in the GAINS model, and are instead prescribed following the RCP8.5 trajectory (Riahi et al., 2011). RCP8.5 methane emissions from international shipping for 2005 are 0.5 Tg $y^{-1}$, accounting for a negligible fraction of total anthropogenic methane emissions. GAINS-derived anthropogenic methane emissions differ from those in the RCP8.5 inventory (http://tntcat.iiasa.ac.at/RcpDb/) by ~ 1 %, indicating good agreement in global magnitude.

**Table 1:** Global annual emissions of reactive non-methane gases and aerosols.

| Pollutant | Sector | Global annual emissions (Tg y$^{-1}$) |
|---|---|---|
| CO | Anthropogenic | 549.8 |
| | Biomass burning | 451.7 |
| | **Total** | **1001.5** |
| NH$_3$ | Anthropogenic | 50.2 |
| | Biomass burning | 10.9 |
| | Ocean | 9.9 |
| | **Total** | **71.0** |
| NO$_X$ (TgN y$^{-1}$) | Anthropogenic | 36.6 |
| | Biomass burning | 5.3 |
| | Lightning [a] | 7.0 |
| | Soil | 2.7 |
| | **Total** | **51.6** |
| SO$_2$ | Anthropogenic | 116.7 |
| | Biomass burning | 3.8 |
| | Volcano | 25.2 |
| | **Total** | **145.7** |
| NMVOC | Anthropogenic | 80.2 |
| | Biomass burning | 49.0 |
| | Vegetation | 41.7 |
| | **Total** | **170.9** |
| BC | Anthropogenic | 6.0 |
| | Biomass burning | 3.6 |
| | **Total** | **9.6** |
| OC | Anthropogenic | 13.7 |
| | Biomass burning | 32.1 |
| | **Total** | **45.8** |
| Isoprene (TgC y$^{-1}$) | **Vegetation [a]** | **340.7** |
| Monoterpenes (TgC y$^{-1}$) | **Vegetation [a]** | **91.3** |
| DMS | **Ocean [a]** | **53.0** |

a) During a simulation, the emission magnitudes of the interactive sectors exhibit interannual variability. The value listed for the interactive emissions is the average calculated over 10 model years. The standard deviation over 10 model years is: 0.08 TgN y$^{-1}$ for lightning NO$_X$; 0.56 Tg y$^{-1}$ for DMS; 4.9 TgC y$^{-1}$ for isoprene; and 1.8 TgC y$^{-1}$ for monoterpenes.

**Table 2:** Global methane emissions and soil sink for 2005.

| Sector | Global annual flux (Tg CH$_4$ y$^{-1}$) |
|---|---|
| Anthropogenic | 325.6 |
| Biomass burning | 24.9 |
| Termites | 6.0 |
| Lakes | 10.0 |
| Terrestrial geological | 20.0 |
| Marine | 5.0 |
| Wetlands | 140.3 |
| Total emissions | 531.8 |
| Soil absorption | -60.0 (uptake) |

Fung et al. (1991) geographically distributed annual methane emissions from termites based on habitat distribution information. Here, the Fung et al. (1991) spatial distribution of the methane emissions from termites is retained, and the global annual flux is scaled to 6 Tg y$^{-1}$, which is the first quartile of the range of published estimates reported both by a recent review (Kirschke et al., 2013) and by the *Fifth Assessment Report* of the Intergovernmental Panel on Climate Change (Ciais et al., 2013). The assigned value is close in magnitude to that suggested by a recent estimate (9 Tg y$^{-1}$, range: 3–15 Tg

y$^{-1}$) that was determined by upscaling ecosystem-specific emission factors (Saunois et al., 2016).

An assessment of the methane budget by the U.S. Environmental Protection Agency (EPA) notes that various inventories might differentially apportion emissions to related source categories, such as for wetland and lake sources or for the various terrestrial and oceanic sources (e.g., gas hydrate, in situ ocean, estuarine, and geological sources; EPA, 2010). Conservative

estimates of the ocean, freshwater, and geological sources are applied to the inventory created here to avoid over counting methane emissions from these categories since different literature references were used to assign the fluxes for these sources. For example, the lake source in this inventory is assigned as 10 Tg y$^{-1}$, evenly distributed over global lake area, which is the lower end of the range (10–50 Tg y$^{-1}$) of published estimates that have been collated by the EPA assessment (EPA, 2010).

Based on published estimates, the EPA assessment reports an ocean methane source in the range of 2.3–15.6 Tg y$^{-1}$, but notes that some of this methane source is likely geological or hydrates (EPA, 2010). The combined ocean plus estuarine source in this inventory is 5 Tg y$^{-1}$, corresponding roughly to the first quartile of the suggested range. The marine methane flux is evenly divided over the global ocean.

A conservative terrestrial geological source of 20 Tg y$^{-1}$ is assigned. Owing to the very large uncertainty in spatial and temporal placement of the fluxes (Etiope et al., 2008), the terrestrial geological component is evenly divided over the Earth's land surface in this inventory. Recent isotopic analyses suggest that the total geological source assigned here might be

underestimated (Schwietzke et al., 2016). The total fossil fraction of methane emissions in the inventory developed here is ~ 31 %, including industrial fossil fuel use, terrestrial geological, and oceanic sources. Based on their reported sector-mean emissions, the total fossil fraction for the period 2003–2013 from the recent Schwietzke et al. (2016) analysis is calculated as ~ 33 %. Their inventory represents an increase in fossil-based methane emissions relative to previous budgets (Schwietzke et

al., 2016). While the fossil fraction for the inventory built here largely matches that of the Schwietzke et al. (2016) analysis, the total magnitude of fossil-based emissions are higher in the Schwietzke et al. (2016) inventory, including geological emissions that are a factor of two stronger than those assigned here. While the gross magnitude of methane emissions is well constrained, substantial uncertainties remain regarding the partitioning of methane emissions among source categories (Rigby et al., 2017; Turner et al., 2017). The interpretation of isotope composition measurements is currently ambiguous and

complex (Turner et al., 2017). Prather and Holmes (2017) have recently suggested new approaches to extract more useful information from existing observations by exploiting spatial patterns.

Some small, uncertain source sectors were not included in the methane budget used in this project. For example, annual methane emissions from permafrost are estimated to be 1 Tg y$^{-1}$ or less (EPA, 2010; Kirschke et al., 2013), but these

estimates are likely upper bounds as they do not account for oxidation of the methane as it travels through the overlying soil to reach the atmosphere (EPA, 2010). No separate permafrost source is included in this inventory.

Using the natural methane flux estimates described here in conjunction with anthropogenic and biomass burning emissions of the short-lived air pollutants from the GAINS and RCP8.5 scenarios, the optimization process employing ModelE2-YIBs

finds that the present-day methane source from wetlands is 140 Tg y$^{-1}$ when a soil sink of 60 Tg y$^{-1}$ is applied. In the Wetland and Wetland CH$_4$ Inter-comparison of Models Project (WETCHIMP) assessment, seven models reported interactive global methane emissions from wetlands (Melton et al., 2013). The multi-model mean ± 1 standard deviation is 190 ± 39 Tg y$^{-1}$ for the WETCHIMP study, with individual models reporting values of 141–264 Tg y$^{-1}$ (Melton et al., 2013). Thus, the wetland methane emission magnitude used in ModelE2-YIBs is 26 % lower than the WETCHIMP multi-model mean, but

almost identically corresponds to the results from one of the individual models, indicating that the prescribed emission magnitude for this highly uncertain sector is reasonable.

The iterative refinement process used to optimize the wetland methane flux was largely a trial-and-error based methodology that made use of literature-derived estimates and surface observations. The wetland methane flux is calculated as a best fit

following prescription of the other fluxes. The baseline wetland methane emissions applied to the optimization process are the methane emissions from bogs and swamps from Fung et al. (1991); the magnitude, spatial distribution, and temporal distribution of these emissions were subsequently modified to varying degrees during the optimization process. At each step of the process, the annual cycle of modeled surface-level methane concentration was compared to observations from the NOAA ESRL measurement network at 50 globally distributed sites (Dlugokencky et al., 2015). The aim of the optimization

process was to minimize the absolute value of the normalized mean bias (NMB) at the largest number of sites. Considering the full set of 50 sites, the final optimized scenario results in NMBs ranging from -1.3 % (model underestimate) to +3.0 % (model overestimate), with a median of +0.4 %. At three quarters of sites, the NMB is between -1 % and +1 %. An evaluation of the simulated atmospheric methane distribution associated with the final optimized emissions inventory, including a comparison to SCIAMACHY methane columns (Schneising et al., 2009), is provided in Sect. 4. Modification of the temporal distribution of wetland methane emissions was guided by both the annual cycles of surface methane concentrations at the 50 NOAA ESRL measurement sites (Dlugokencky et al., 2015) and the annual cycle of wetland methane emissions simulated by the models participating in the WETCHIMP analysis (Melton et al., 2013).

The best fit of modeled atmospheric methane relative to the NOAA ESRL surface methane observations corresponds to the following modification of the baseline wetland methane emissions dataset. First, the baseline wetland methane emissions (extratropical bogs and tropical swamps) from Fung et al. (1991) were scaled to achieve an extratropical emissions fraction of 30 % and a prescribed global emission magnitude of about 130 Tg $CH_4$ $y^{-1}$. A single scaling factor was applied in each grid cell in each month to the emissions from bogs; likewise, a separate single scaling factor was applied in each grid cell in each month to the emissions from swamps. For the WETCHIMP study, the mean extratropical emissions fraction among all participating models is about 30 % (Melton et al., 2013). Secondly, an additional 10 Tg $CH_4$ $y^{-1}$ was added to the wetland methane emissions: (1) 2 Tg $CH_4$ $y^{-1}$ was added to 20°N–40°N over the months March through September; (2) 2 Tg $CH_4$ $y^{-1}$ was added to 0°–20°N over the months May through October; and (3) 6 Tg $CH_4$ $y^{-1}$ was added to 20°S–0° over all months. Finally, the seasonal cycle of the wetland methane emission hotspots in Finland and Russia (50°N–70°N) were adjusted: 0.5 Tg $month^{-1}$ decrease for each of June, July, and August; 0.65 Tg $month^{-1}$ increase in both September and October, and 0.2 Tg $month^{-1}$ increase in November.

The methane soil sink in the ModelE2-YIBs inventory corresponds to the top end of the range suggested by the review of Dutaur and Verchot (2007) but is higher than the magnitude reported in recent reviews (e.g., top-down range: 26–42 Tg $y^{-1}$; bottom-up range: 9–47 Tg $y^{-1}$; Kirschke et al., 2013). The wetland methane emissions are derived as a best fit given the other prescribed emissions, the methane soil sink, and the simulated chemical sink. Applying a weaker soil sink would have resulted in a lower magnitude for the derived wetland methane emissions; applying a stronger soil sink would have resulted in a higher magnitude for the derived wetland methane emissions. The simulated total atmospheric lifetime of methane and the simulated methane mixing ratio in ModelE2-YIBs are well aligned with observation-based estimates (Sect. 4), suggesting that the overall rate of removal of methane is well represented in the model.

The annual cycle of wetland methane emissions is plotted in Fig. 1. Monthly emissions are shown for the same latitudinal zones that are plotted in Melton et al. (2013) for six models participating in the WETCHIMP analysis (their Fig. 6, corresponding to the mean annual cycle for years 1993–2004). Global monthly methane emissions from wetlands range from

7.4–18.2 Tg month$^{-1}$ (Fig. 1). Monthly emissions show little variability from November to April (range: 7.4–9.5 Tg month$^{-1}$), followed by increasing emissions starting in May (12.9 Tg month$^{-1}$). Peak monthly emissions occur in July (18.2 Tg month$^{-1}$). The six WETCHIMP models simulate peak emissions, variously occurring between June and August, of slightly higher magnitude (approximate range for the six models: 20–35 Tg month$^{-1}$; Melton et al., 2013). The annual cycle of emissions for the 40°N–90°N latitudinal band is similar in shape to that for global emissions, with peak monthly emissions likewise occurring in July (9.1 Tg month$^{-1}$; Fig. 1). Monthly emissions for the 20°N–40°N band show little variation throughout the year and are of low magnitude (range: 0.5–0.9 Tg month$^{-1}$; Fig. 1), while the WETCHIMP models generally exhibit a small peak on the order of 5 Tg month$^{-1}$ in this band in the Northern Hemisphere summer (Melton et al., 2013). The 0°–20°N band shows increasing monthly emissions between February and August, followed by declining monthly emissions (Fig. 1). The 20°S–0° band shows the largely opposite cycle, with minimum monthly emissions occurring in August (1.4 Tg month$^{-1}$). Monthly emissions from the tropics, considering 30°S–30°N, are largely consistent throughout the year, ranging from 6.0–8.0 Tg month$^{-1}$.

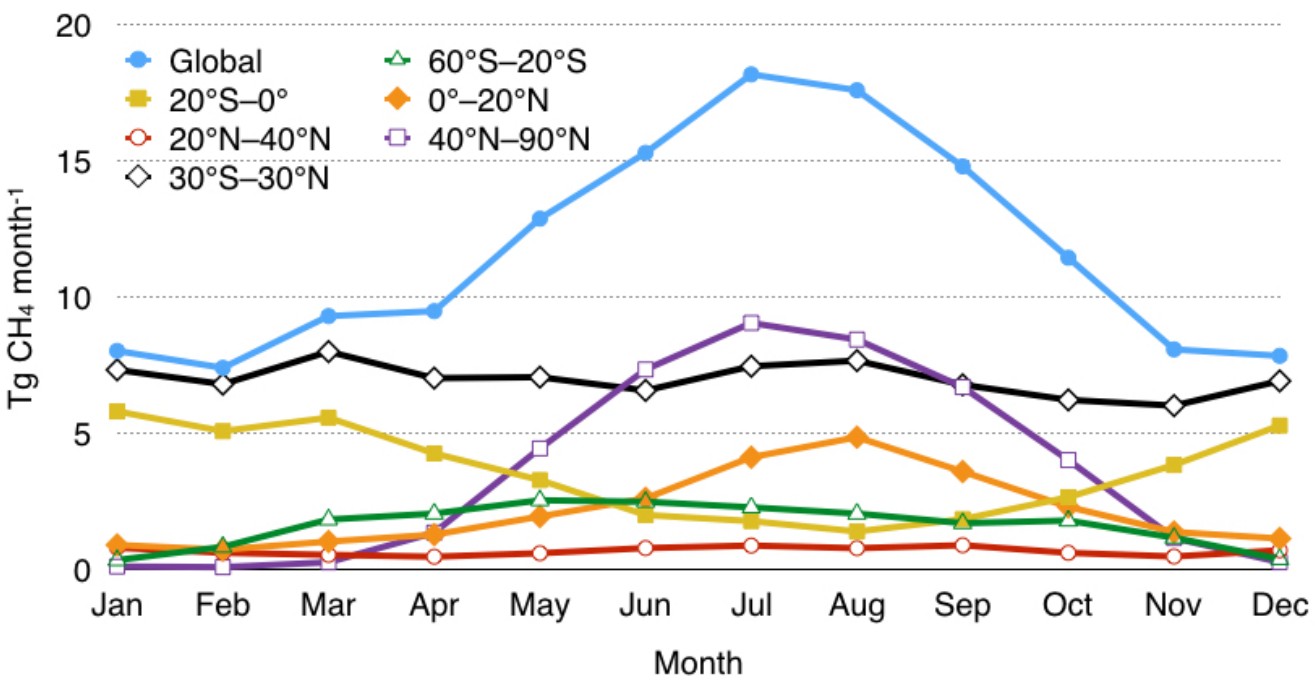

**Figure 1:** Monthly wetland methane emissions (Tg CH$_4$ month$^{-1}$) for several latitudinal bands for the optimized inventory.

The zonal distribution of annual wetland methane emissions is shown in Fig. 2, with emissions aggregated over 2°-latitude bands. Peak annual emissions occur near the equator, similar to the WETCHIMP multi-model mean (Melton et al., 2013, their Fig. 5, although shown in 3°-latitude bands). In the Southern Hemisphere, the optimized wetland methane inventory exhibits smaller secondary peaks near 15°S and 30°S. The WETCHIMP multi-model mean likewise exhibits regional peaks

in these locations, but the magnitude of the peak at 30°S relative to the peak at the equator is stronger in the optimized inventory than in the WETCHIMP analysis. Like the WETCHIMP multi-model mean, the optimized wetland emissions inventory shows a wide secondary peak centered around 55°N. The secondary peak at 10°N is also seen in the WETCHIMP multi-model mean; in the optimized inventory, this peak exhibits a stronger magnitude relative to the main peak at the equator than occurs in the WETCHIMP analysis. The spatial distributions of the monthly wetland methane emissions are

shown in Fig. S1, and the gridded optimized monthly wetland methane emissions data are provided in the Supplementary Information.

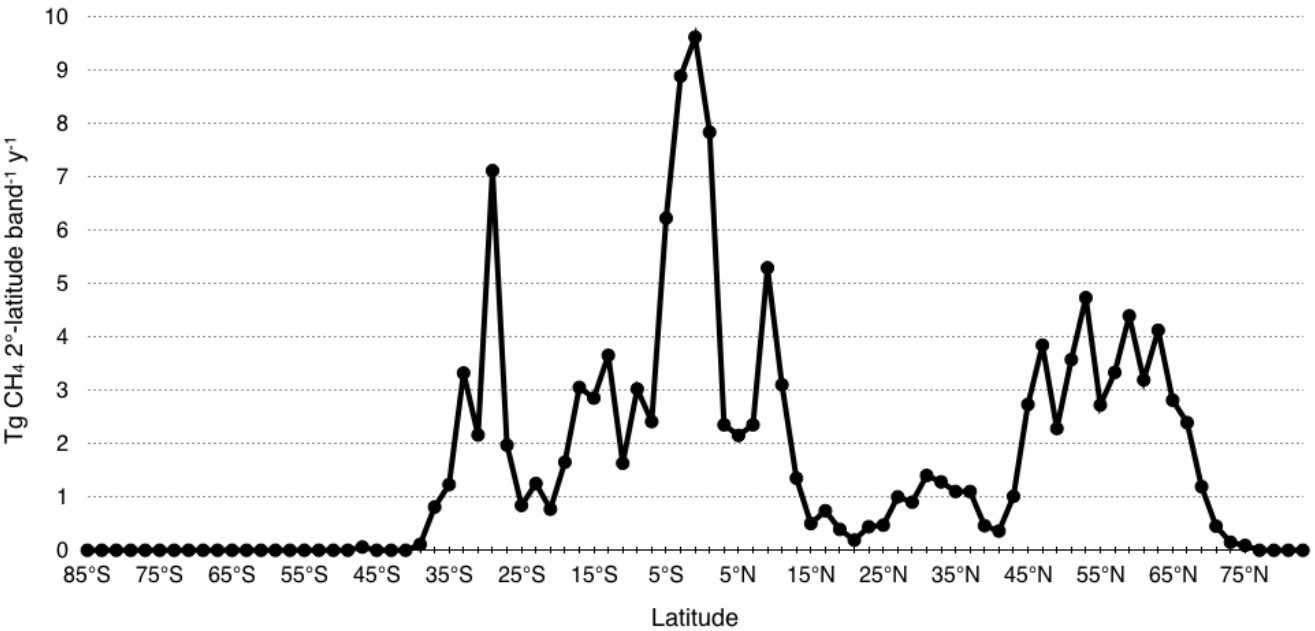

**Figure 2:** Annual zonally summed wetland methane emissions (Tg CH$_4$ 2°-latitude band$^{-1}$ y$^{-1}$) for the optimized inventory.

Total annual methane emissions from all non-oceanic sources are shown in Table 3 for 14 regions. Regional definitions follow Saunois et al. (2016). In their Table 4, Saunois et al. (2016) provide estimates of annual methane emissions (means

for 2000–2009) for the same 14 regions, including both best estimates and ranges resulting from a set of inversions. The

regional methane emissions from the optimized inventory fall within the suggested range of Saunois et al. (2016) for nine regions: temperate South America, tropical South America, central North America, boreal North America, southern Africa, northern Africa, Europe, China, and Oceania. For two other regions (contiguous USA and India), the emissions fall within 1–2 Tg y$^{-1}$ of the suggested range. Emissions in Southeast Asia from the optimized inventory are slightly lower than the

5   range of 54–84 Tg y$^{-1}$ suggested by Saunois et al. (2016). The optimized inventory exhibits emissions that are higher than the suggested ranges of Saunois et al. (2016) for two regions: (1) Russia (suggested range: 32–44 Tg y$^{-1}$) and (2) Central Eurasia and Japan (suggested range: 38–51 Tg y$^{-1}$). For both regions, the strong emissions in the inventory applied here are associated with strong energy sector emissions and, in the case of Russia, strong wetland emissions. Comparison of simulated column-average methane concentrations with those from SCIAMACHY (Sect. 4.2) shows model underestimates

10   on the order of 2% in these regions, which is typical of model underestimates in other regions. The global distributions of annual methane emissions by source category are shown in Fig. S2. The total emission magnitude of methane for 2005 in the ModelE2-YIBs inventory is 532 Tg y$^{-1}$ (Table 2), which corresponds well to the top-down estimate (548 Tg y$^{-1}$, range: 526–569 Tg y$^{-1}$) reported by the Kirschke et al. (2013) review and is only slightly outside of the range from the top-down estimate (552 Tg y$^{-1}$, range: 535–566 Tg y$^{-1}$) reported by the more recent Saunois et al. (2016) review.

**Table 3:** Regional annual methane emissions from non-oceanic sources (Tg y$^{-1}$). Regional definitions follow Saunois et al. (2016).

| Region | Annual methane emissions (Tg y$^{-1}$) |
| --- | --- |
| Temperate South America | 23.0 |
| Tropical South America | 70.4 |
| Central North America | 12.1 |
| Contiguous USA | 37.0 |
| Boreal North America | 17.7 |
| Southern Africa | 37.8 |
| Northern Africa | 38.4 |
| Europe | 30.6 |
| Russia | 60.7 |
| Central Eurasia and Japan | 57.2 |
| China | 50.5 |
| India | 26.3 |
| Southeast Asia | 47.4 |
| Oceania | 17.1 |

## 4 Simulated methane in ModelE2-YIBs

The annual-mean mixing ratio of surface-level methane for E2005 is plotted in Fig. 3. The global map indicates strong spatial heterogeneity, with local surface concentrations ranging from 1664 to 2198 ppbv. Source regions with strong methane emissions are readily apparent, such as parts of Russia, South America, and central Africa (large wetland sources) and the Middle East and China (large anthropogenic sources, including agricultural sources in the case of China). The model output indicates a large inter-hemispheric difference in surface-level methane concentrations, driven by comparatively strong emissions in the Northern Hemisphere (NH) relative to the Southern Hemisphere (SH).

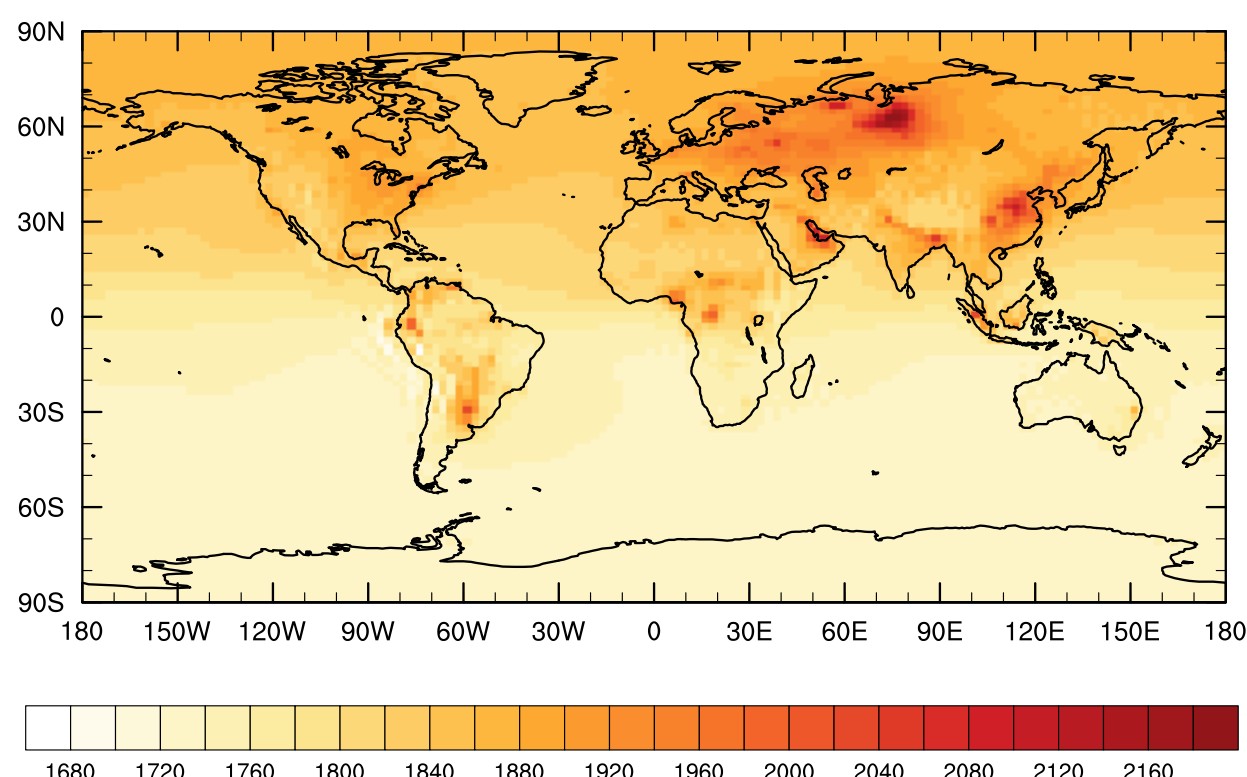

**Figure 3:** Simulated annual-mean surface methane mixing ratio (ppbv) for year 2005.

Based on application of the year 2005 emission inventory to ModelE2-YIBs, the simulated hemispheric-mean surface methane mixing ratios are 1746 ppbv for the SH and 1841 ppbv for the NH. The simulated global-mean surface methane concentration of 1793 ppbv is only 1.1 % higher than the observed value for 2005 derived from the NOAA ESRL global air-sampling network (Dlugokencky et al., 2015). The small model overestimate is only slightly higher in the methane-emissions-rich NH (+1.3 %) than in the comparatively methane-emissions-poor SH (+0.9 %). Both the model and the NOAA ESRL measurements indicate an inter-hemispheric ratio (NH:SH) of 1.05. This comparison indicates that the broad pattern of surface methane concentration simulated by the model is realistic.

A spatially explicit validation of the simulated atmospheric methane distribution is achieved through comparison of the E2005 output with (1) NOAA ESRL surface measurements from 50 globally distributed stations (Dlugokencky et al., 2015), described in Sect. 4.1, and (2) methane columns derived from the SCIAMACHY instrument aboard the ENVISAT satellite (Schneising et al., 2009), described in Sect. 4.2.

## 4.1 Comparison with surface measurements

The model–measurement comparison making use of the NOAA ESRL surface measurements (Dlugokencky et al., 2015) is performed for each measurement station that has at least one data point available per calendar month for the period 2001–2005. The locations of the 50 measurement stations that fulfill this criterion are identified on the map in Fig. S3. These 50 measurement stations collectively span latitudes extending from 90º S to 82.5º N. Roughly three-quarters of the measurement stations are located in the Northern Hemisphere. There is a dearth of land-based measurement sites located in South America, Africa, and Australia. For each measurement site, the analysis uses all monthly observations available for the period 2001–2005 along with the E2005 output for the overlapping model grid cell.

A latitudinal gradient in the annual-mean surface methane mixing ratio is evident in both the observations and model results (Fig. 4). The relative difference between model and observation ranges from a model underestimate of 1.3 % in Moody, Texas, (31.3º N, 97.3º W) to a model overestimate of 3.0 % on the Tae-ahn Peninsula (36.7º N, 128.1º E). The simulated methane concentration is within 1 % (i.e., -1 % to +1 %) of the measured value at 76 % of locations. Only three sites exhibit an overestimate > 2 %. Considering all 50 sites, the average relative difference between model and observations is a model overestimate of 0.5 % (median = 0.4 %), indicating that the model skillfully simulates annual-mean surface methane mixing ratios.

Figure 5 shows the annual cycles for the 50 measurement stations. The individual panels also report the normalized mean bias (NMB; %) calculated using monthly means for each measurement location; mathematically, the NMB based on monthly means is equal to the relative difference (%) in annual means. At most measurement sites, the simulated annual cycle of surface methane largely mimics the observed cycle. In the Southern Hemisphere middle to high latitudes, the model

accurately reproduces the measured austral winter methane maximum. At these sites, the model overestimates the austral summer minimum by ~ 1 %, suggesting that the model slightly underestimates summertime chemical loss. The model also overestimates boreal summer methane minimums at the Northern Hemisphere high latitude sites (e.g., Summit station), which is similarly likely due to a model underestimate in summertime chemical loss. The model–measurement differences in

annual cycles might also be associated with the temporal and spatial assumptions made in the prescribed methane emissions inventory. The model fails to capture the annual cycle at a few locations, notably Pallas-Sammaltunturi in Finland; Barrow in Alaska, USA; and Ulaan Uul in Mongolia. The poor correlation between observed and modeled cycles for this limited set of stations is likely associated with localized sources and sinks near the measurement sites that are not accounted for in the large-scale model. Based on interactive methane simulations with the HadGEM2 chemistry–climate model, Hayman et al.

(2014) likewise found model–measurement discrepancies in the annual cycles at these and other sites, finding that, in their simulations, the Barrow and Pallas-Sammaltunturi sites are strongly influenced by emissions from wetlands, while the Ulaan Uul site is influenced by other non-wetland emission sources.

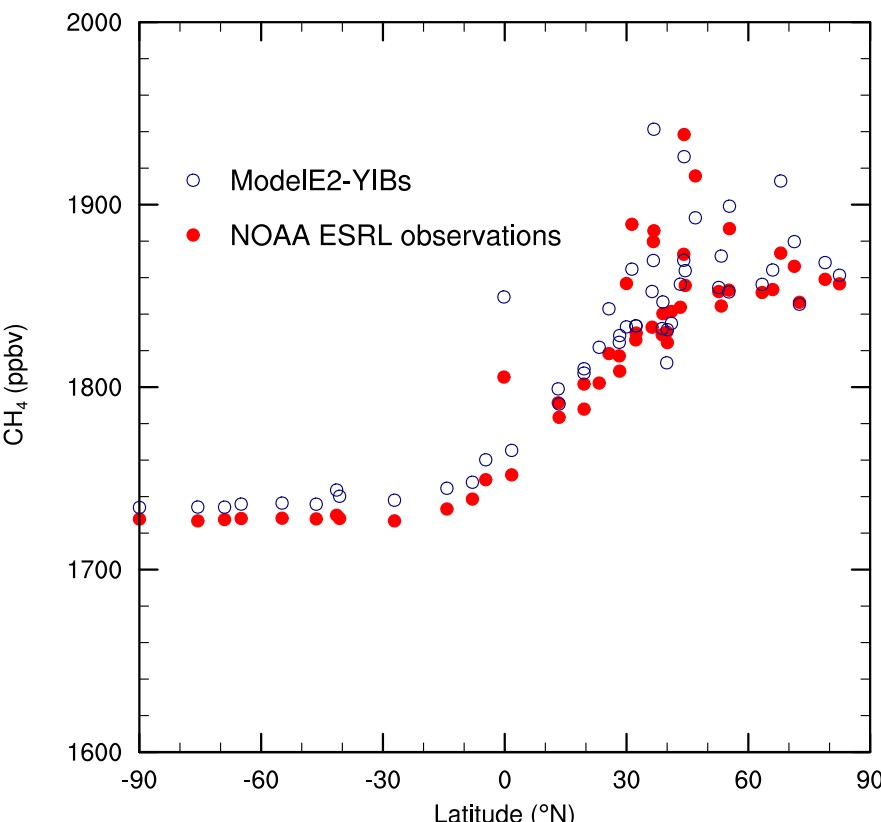

**Figure 4:** Annual-mean surface methane concentration (ppbv) at 50 locations for both the E2005 simulation and the NOAA ESRL measurements.

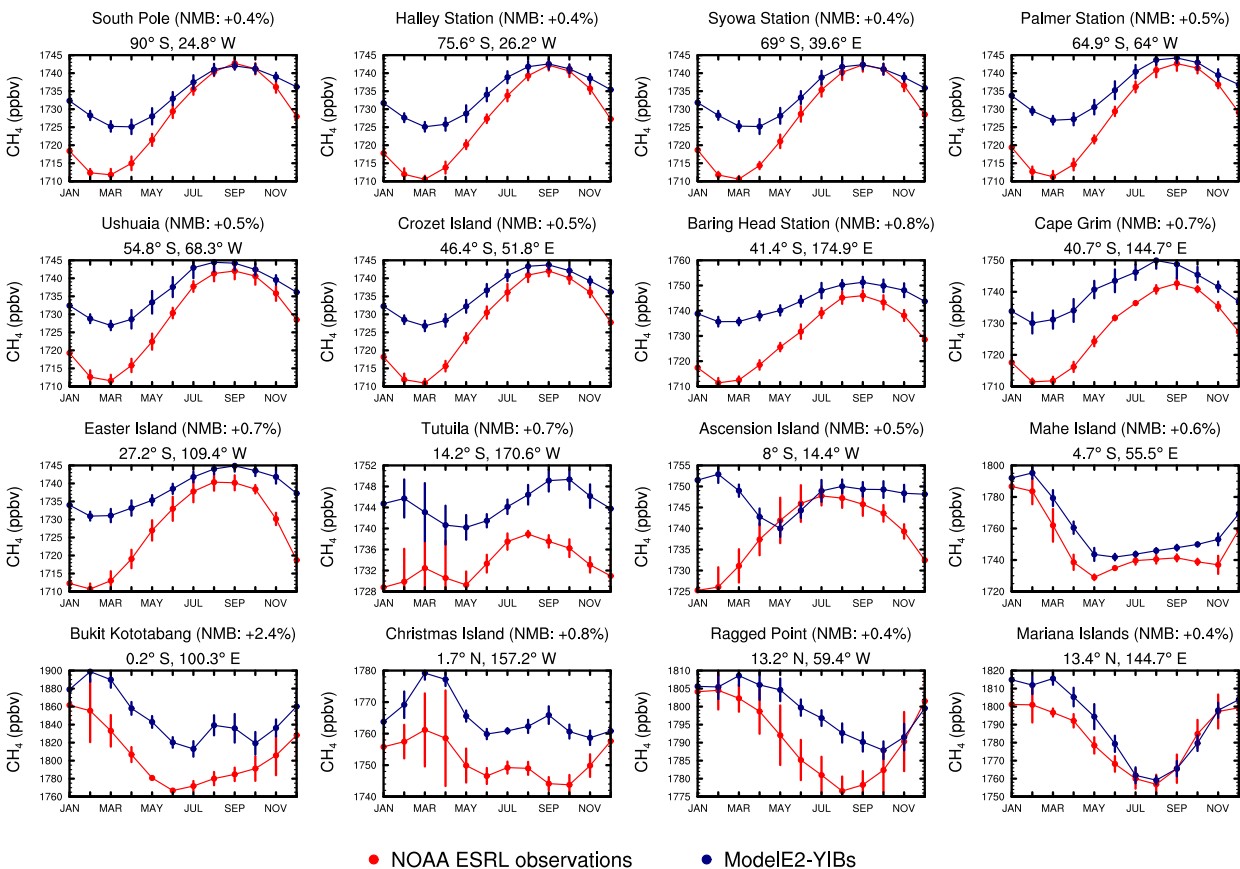

**Figure 5:** Annual cycle of surface methane concentration (ppbv) at 50 locations for both the E2005 simulation and the NOAA ESRL measurements. The filled circles represent monthly means, and the vertical bars represent ± 1 standard deviation. The scale varies by panel. The normalized mean bias (%) calculated using monthly means is indicated in the panel titles.

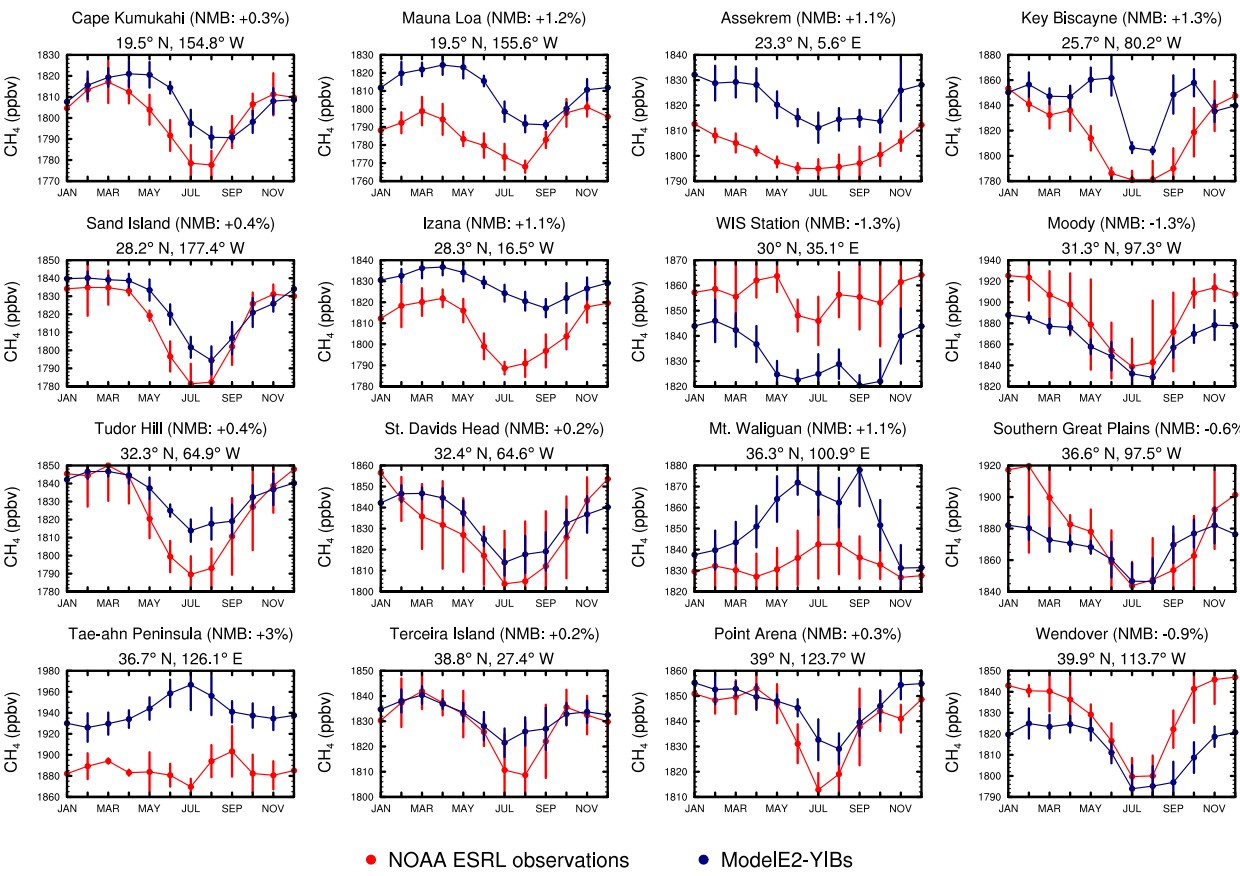

**Figure 5:** Continued.

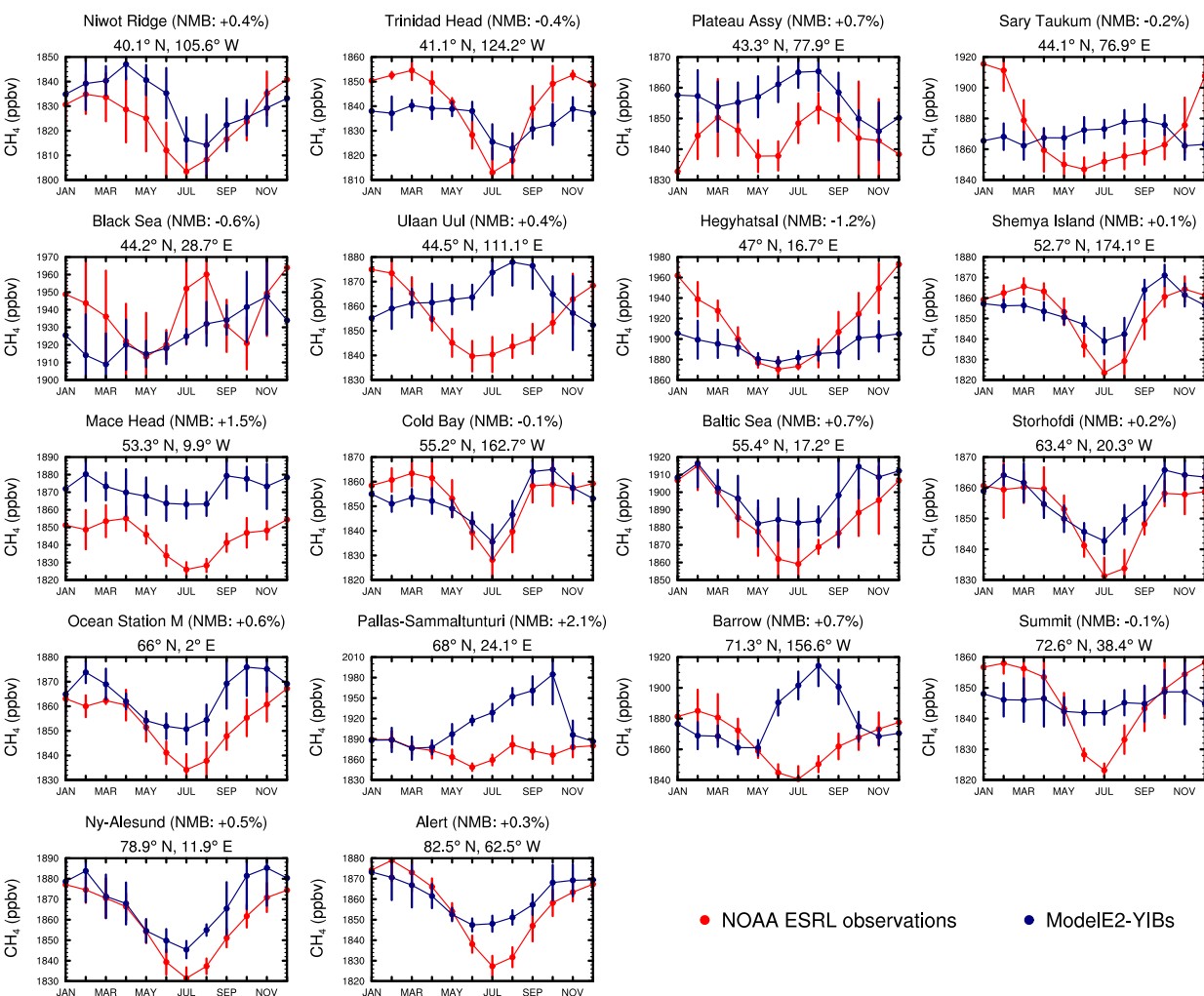

**Figure 5:** Continued.

## 4.2 Comparison with satellite retrievals

SCIAMACHY methane columns are available at near-global coverage (Schneising et al., 2009), providing a means to evaluate model performance in regions not covered by the more limited NOAA ESRL surface measurement network. Comparison of modeled methane with SCIAMACHY data provides an independent post-optimization evaluation. The relative differences in annual column-averaged methane mixing ratios for E2005 and SCIAMACHY are plotted in Fig. 6. The SCIAMACHY instrument experienced degraded detector performance beginning in November 2005 (Schneising et al., 2009); as such, the model validation using SCIAMACHY-derived methane columns makes use of all satellite observations

available for the period November 2002 to October 2005 (i.e., 3 years of observations for each calendar month). To account for the altitude sensitivity of the satellite retrievals, the model data were sampled using the SCIAMACHY averaging kernels and a priori mole fractions (Schneising et al., 2009). In each model grid cell, the simulated annual-mean mixing ratio was calculated using only the monthly means corresponding to the calendar months for which SCIAMACHY has available data.

Ninety-five percent of grid cells with data exhibit a model underestimate in column-averaged methane, indicating that the total methane source strength in the model is slightly too weak or the methane sink strength is slightly too strong. The model underestimate is slight in most grid cells: 83 % of grid cells with data exhibit an underestimate of < 3 %. The global-mean relative difference in methane columns is a model underestimate of 1.7 %. Both hemispheres exhibit an identical model underestimate (1.7 %), indicating relative spatial uniformity in model performance. NOAA ESRL surface measurement stations are largely absent from South America, Africa, and Australia (Fig. S3). Comparison of the modeled methane columns with SCIAMACHY retrievals indicates that the model underestimate on these continents is ~ 1 to 3 % in most locations, which is equivalent to the underestimates simulated for North America, Europe, and most of Asia outside of the Tibetan Plateau. Using interactive methane simulations in the HadGEM2 chemistry–climate model, Hayman et al. (2014) likewise found that the model underestimated column-averaged methane concentrations relative to SCIAMACHY observations due to simulated methane concentrations that decreased too rapidly with increasing altitude. The HadGEM2 simulations applied an explicit methane loss term to represent stratospheric methane oxidation (Hayman et al., 2014), while ModelE2 uses fully coupled dynamic stratospheric chemistry (e.g., Shindell et al., 2006).

The model slightly overestimates annual-mean surface methane at 80 % of the NOAA ESRL measurement locations and underestimates column-averaged methane at most locations on the globe. This mis-match could indicate that the principal chemical sink of methane – reaction with OH – is slightly too strong in the model outside of the surface layer, or it could indicate potential issues with the transport mixing rate of methane in the free troposphere and stratosphere. Future work with other vertically resolved satellite data products may help unravel the chemical and/or dynamical causes. Overall, the model shows good agreement with measured methane mixing ratios, providing confidence in its ability to simulate the principal chemical and dynamical processes that affect methane in the atmosphere.

## 4.3 Methane lifetime

Further evidence of the model's skill in capturing methane-relevant processes is found through the close agreement of methane lifetime in the model with that derived from observations. The chemical lifetime of methane in E2005 is $10.4 \pm 0.1$ years, which is nearly identical to the present-day methane chemical lifetime against OH of $10.6 \pm 0.4$ years that was derived from OH estimates based on methyl chloroform observations (Rigby et al., 2013). The methane chemical lifetime in the model is only slightly shorter than – but well within the 1 standard deviation range of – a second observation-based estimate

that is likewise based on methyl chloroform loss to OH: 11.2 ± 1.3 years for 2010 (Prather et al., 2012). The total lifetime of methane in E2005, taking into account both chemical loss and the soil sink, is 9.2 ± 0.04 years. This closely matches the present-day methyl chloroform-based estimates of total methane lifetime of 9.7 ± 0.4 years (Rigby et al., 2013) and 9.1 ± 0.9 years (Prather et al., 2012), derivation of which makes use of estimates of the loss rates for the other minor methyl

5    chloroform and methane sinks. Importantly, the close agreement between the modeled and observation-based methane lifetimes is a strong indicator that the model appropriately captures the processes that control atmospheric methane.

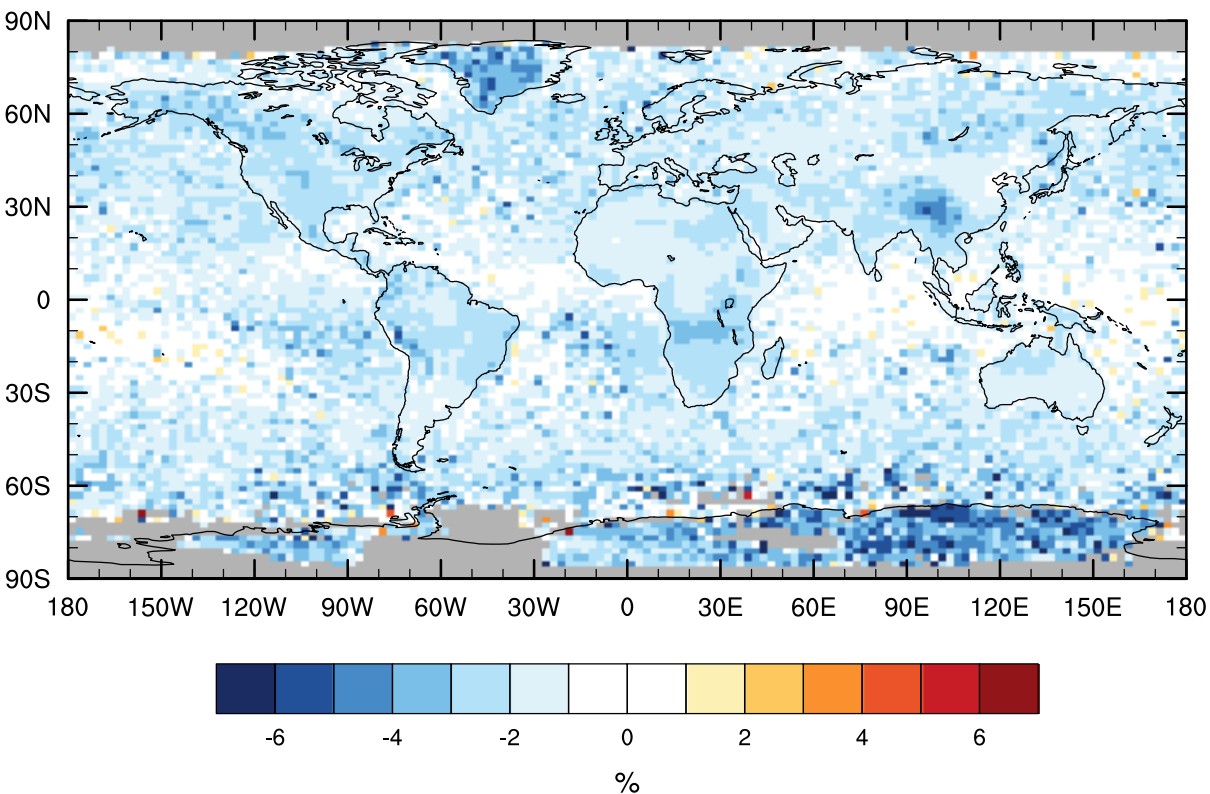

10   **Figure 6:** Relative difference (%) between simulated (E2005) and SCIAMACHY annual column-averaged methane concentrations. Relative difference = 100 × (model – satellite)/satellite. Range = -11.2 to +7.1%.

## 5 Simulated ozone in ModelE2-YIBs

The simulated tropospheric ozone burden for E2005 is 353 ± 1.5 Tg, which falls well within the range (302–378 Tg, for year 2000) reported for the 15 global models that participated in the Atmospheric Chemistry and Climate Model Intercomparison Project (ACCMIP; Young et al., 2013) and is only 5 % higher than the ACCMIP multi-model mean (337 ± 23 Tg),

5 indicating good agreement with other global models. The magnitudes of the simulated annual ozone fluxes are likewise supported by the results of the ACCMIP study, although only six ACCMIP models report ozone flux magnitudes for year 2000 (Young et al., 2013). The simulated magnitude of the annual net flux of ozone from the stratosphere to the troposphere (452 ± 16 Tg $y^{-1}$) falls within the ACCMIP range (401–663 Tg $y^{-1}$) as does the simulated magnitude of net chemical production (907 ± 17 Tg $y^{-1}$ for E2005; ACCMIP range: 239–939 Tg $y^{-1}$). The simulated annual ozone dry deposition flux

10 (1359 ± 5.7 Tg $y^{-1}$) is only 0.7 % higher than the top end of the ACCMIP range (687–1350 Tg $y^{-1}$). Overall, the simulated ozone budget for E2005 shows good agreement with those reported by the global models that participated in ACCMIP.

Validation of the simulated ozone concentrations for E2005 is achieved through comparison with an ozonesonde climatology (Tilmes et al., 2012) that provides ozone concentrations at 26 pressures for 41 measurement stations. The Tilmes et al.

15 (2012) climatology is based on measurements from the period 1995–2011, while the E2005 simulation is roughly representative of year 2005. Ozone concentrations may have changed in some regions over the 1995–2011 era (e.g., Cooper et al., 2014); thus, the ozonesonde climatology is used only to provide validation that the model captures the broad patterns of the global distribution of ozone at the turn of the century. The distribution of measurement sites is shown in Fig. S4. Roughly half of the sites are located in either North America or Europe; the other continents are poorly represented, although

20 there is significant coverage at remote tropical sites.

Figure 7 plots the annual-mean ozone mixing ratios from the ozonesonde climatology and simulation E2005, with comparisons shown for four pressures. The data points are arranged according to the latitudes of the measurement stations. The simulated ozone data correspond to the grid cells that overlap the individual measurement stations. In the lower

25 troposphere (800 hPa), there is better agreement between modeled and measured ozone at sites in the Southern Hemisphere and in the Northern Hemisphere tropics than at sites in the Northern Hemisphere mid-latitudes. In the Northern Hemisphere mid- and high-latitudes, the model shows a positive bias relative to observations. Better agreement between the climatology and the E2005 simulation can be expected for the less polluted sites. At the more polluted Northern Hemisphere mid-latitudes, strict agreement cannot be expected between the 17-year climatology and the simulated year 2005 that falls toward

30 the tail end of the climatological period. Nonetheless, for the most part, both model and measurements show higher ozone concentrations at 800 hPa in the Northern Hemisphere mid-latitudes than in the Southern Hemisphere.

The NMB of modeled ozone at 800 hPa relative to the climatology ranges from -17.9 to +41.4 % for the set of 41 sites (Table 4). All NMB calculations are based on monthly-mean ozone concentrations. The model likewise exhibits a positive bias at most Northern Hemisphere sites in the middle troposphere (500 hPa, Fig. 7). At many of the Northern Hemisphere sites, the model exhibits an NMB of smaller magnitude at 200 hPa than at either 500 or 800 hPa. At 90 hPa, the model

5    underestimates stratospheric ozone relative to the climatology in the polar regions of both hemispheres.

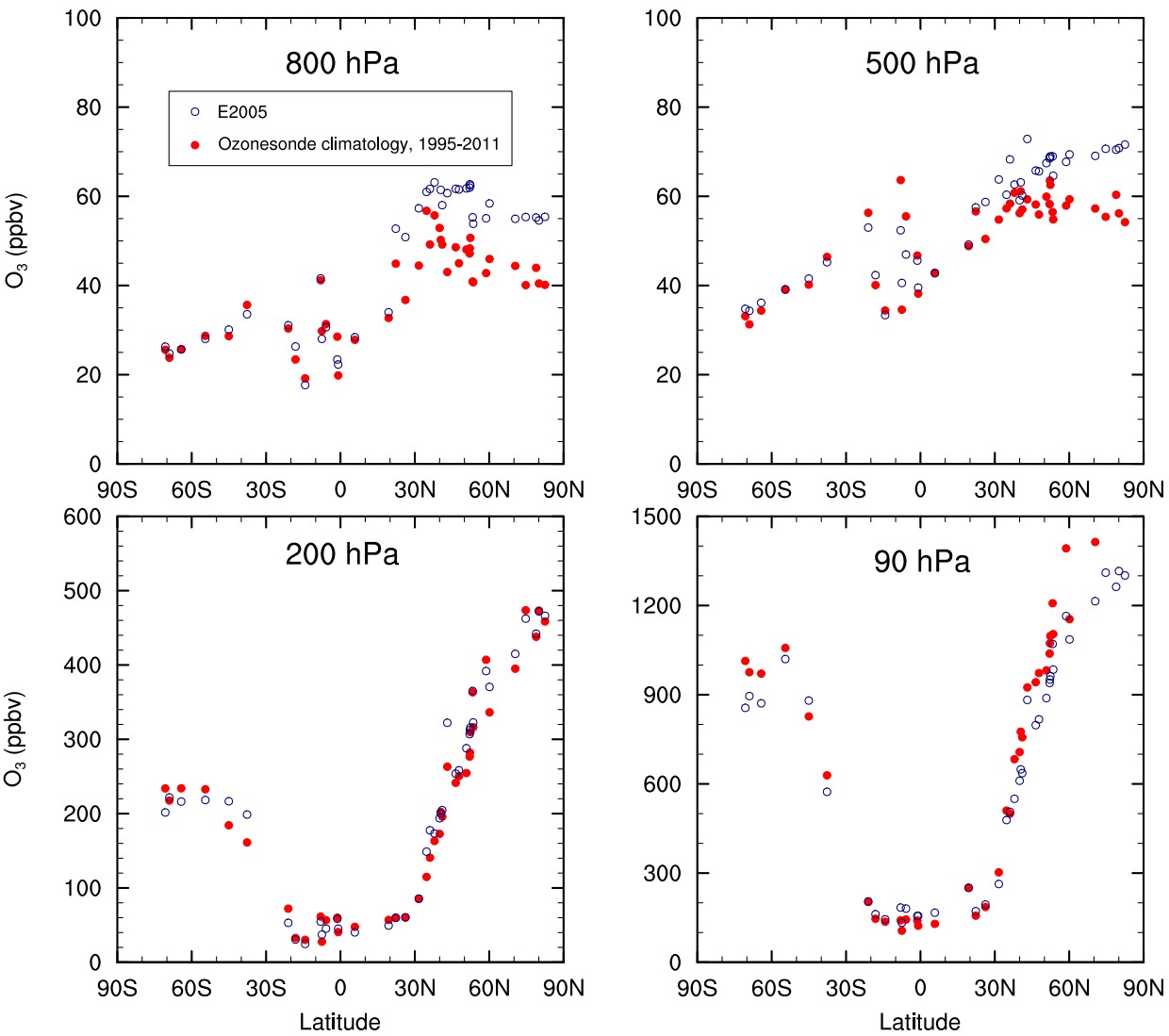

**Figure 7:** Annual-mean ozone concentration (ppbv) at 41 locations for four pressures for both the E2005 simulation and the

10   Tilmes et al. (2012) ozonesonde climatology.

**Table 4:** NMB (%) of ozone mixing ratios for the E2005 simulation relative to the Tilmes et al. (2012) ozonesonde climatology.

| Pressure (hPa) | Minimum | Maximum | Median | Mean |
|---|---|---|---|---|
| 800 | -17.9 | +41.1 | +20.1 | +16.9 |
| 500 | -17.7 | +32.1 | +8.3 | +9.2 |
| 200 | -26.5 | +34.7 | +1.7 | +2.6 |
| 90 | -20.9 | +30.3 | -9.5 | -3.9 |

For each measurement location and pressure, NMB is calculated using monthly means. Indicated for each pressure is the minimum, maximum, median, and mean NMB from the full suite of 41 stations.

## 6 Conclusions

The results of the optimization process using atmospheric modeling indicate global annual methane emissions of 140 Tg $CH_4$ $y^{-1}$ from wetlands; this derivation assumes accurate representation of the other methane fluxes and atmospheric chemical loss in the model. The global annual methane emissions magnitude from all natural sources is 181 Tg $CH_4$ $y^{-1}$. Overall, the total global annual methane emissions magnitude in E2005 is 532 Tg $CH_4$ $y^{-1}$, taking into account the natural flux inventory, anthropogenic emissions derived from the GAINS integrated assessment model (Amann et al., 2011), and biomass burning and international shipping emissions from the RCP8.5 scenario (Riahi et al., 2011). The total emission magnitude falls well within the range reported by a recent review (Kirschke et al., 2013). Comparison with multiple observational datasets indicates close agreement between measured and modeled methane lifetime and atmospheric distribution. The good model–measurement agreement indicates that the interactive chemistry scheme in the ModelE2-YIBs global chemistry–climate model, when forced with the updated natural methane flux inventory, appropriately represents the principal chemical and physical processes that affect atmospheric methane, providing confidence in the model's ability to appropriately capture the methane response to perturbations in emissions of both methane and other short-lived air pollutants. The improved methane scheme is currently being applied to time-slice chemistry–climate simulations to quantify the methane response and concomitant radiative forcing associated with perturbations in anthropogenic methane emissions. The gridded natural methane fluxes associated with the optimized methane scheme in ModelE2-YIBs are provided in the Supplemental Information. This dataset can serve as a useful starting point for optimization of the interactive methane schemes in other atmospheric models. Starting with a reasonable approximation of prescribed methane fluxes can reduce the computational power and time needed for optimization in other models, potentially prompting more widespread use of interactive methane schemes in global modeling. The optimized methane inventory developed in this study additionally

serves as a useful starting point for a potential follow-up study aimed at optimization for transient simulations, in which the prescribed methane emissions evolve over time.

## Code and data availability

The source code for the site-level YIBs model version 1.0 is available at https://github.com/YIBS01/YIBS_site. The GISS ModelE2 source code can be obtained from NASA GISS (https://www.giss.nasa.gov/tools/modelE/). Included as supplemental information are the gridded natural methane fluxes and the numerical model output used to make the figures. Gridded files of natural methane fluxes associated with the Fung et al. (1991) dataset were obtained from NASA GISS (data.giss.nasa.gov/ch4_fung). Column-averaged methane concentrations from SCIAMACHY were obtained from the University of Bremen (iup.uni-bremen.de/sciamachy/NIR_NADIR_WFM_DOAS/products/). Other data used as model input or for analysis of model output are listed in the references.

## Author contribution

K.H. and N.U. designed the study. K.H. and Y.Z. performed the model simulations. K.H. analyzed the model output and satellite data. K.H. prepared the manuscript with revisions from all co-authors.

## Competing interests

The authors declare that they have no conflict of interest.

## Acknowledgements

This project was supported in part by the facilities and staff of the Yale University Faculty of Arts and Sciences High Performance Computing Center. The authors thank Vaishali Naik for providing programming code to read the pre-processed methane surface measurement data; Chris Heyes and Zbigniew Klimont for providing access to and assistance with the GAINS-derived anthropogenic emissions inventory; and Greg Faluvegi for providing guidance on running interactive methane simulations with ModelE2.

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
