# Peer review of "Advances in representing interactive methane in ModelE2-YIBs (version 1.1)"

_Geoscientific Model Development, 2018_

## Referee Comment (RC1) · Anonymous Referee #1 · 2 Jun 2018

Harper and coauthors present a global atmospheric chemistry-climate model with methane emissions. The paper documents the emissions of methane and other compounds, then evaluates the simulated concentrations of methane and ozone against observations. The methods are reasonable and the comparison to observations is sufficient to show that the model appears to be performing competently. The paper is written clearly. Some methods need greater explanation and discussion, which can be accomplished with modest revisions.

The model construction and budget analysis are based on the assumption that atmospheric methane was in steady state in 2005. Although the atmospheric methane concentrations were approximately stable during 2000-2007, as the authors say on p7, atmospheric methane may not have been in steady state at that time because

emissions and OH may have been changing (Rigby et al. 2017; Turner et al., 2017). The steady state assumption, its limitations, and implications for model interpretation should be discussed.

In the abstract and conclusions, the emissions magnitude and especially its partitioning into natural sources are stated too confidently and simply. These estimates assume that the prescribed anthropogenic emission inventory and the simulated CH4 loss are correct. Any error in these other budget terms would alter the authors' estimate of natural emissions. The emissions values should be presented as a best fit within the context of the other model assumptions.

The paper needs greater detail about how the natural methane emissions were optimized. The general approach is described a bit in Section 2, but lacks detail for a reader to attempt to reproduce it. I suggest providing this greater detail in Section 3. What observations were used in the optimization? Was it a formal optimization of some cost function or ad hoc trial and error with visual comparison? I would expect that the optimal emissions would produce an unbiased global mean, but Section 4.1 reports and Figs 2 and 3 show that the model is systematically higher than observations at almost all sites.

In the abstract and elsewhere, 1% model bias against surface observations is acceptable, but not excellent. It may be comparable to the performance of other models, but it is one-fifth of the interhemispheric ratio NH/SH: 1.05. For a well-mixed gas like methane, a 1% model error after optimization is substantial.

The supplement contains data in Excel xlsx format. I recommend an open source file format readable by free software, but I defer to the editor on whether this is required.

Minor comments

P2L19: CH4 is also oxidized by O(1D) in the stratosphere.

P5L13. Is version 1.1 a past model version or the new version described by this paper?

[Figure]

P6L21. Were the LLGHG concentrations prescribed at the surface or also elsewhere?

P10L9 The work of Turner et al. is slightly misrepresented. The gross magnitude of methane emissions are well constrained, with uncertainty of 10% or less in the global total (Turner et al., 2017; also Prather et al., 2012). Turner et al. (2017) and also Rigby et al. (2017) showed that showed that observations poorly constrain partitioning and small but important trends in this total, although see Prather and Holmes (2017) for ways that exploiting spatial patterns could extract more information from existing observations.

P17L25. I believe the CH4 lifetime estimate by Rigby et al. (2013) should supersede Prinn et al. (2005), although the values are similar.

References

Prather, M. J., & Holmes, C. D. (2017). Overexplaining or underexplaining methane's role in climate change. Proceedings of the National Academy of Sciences, 114(21), 5324–5326. http://doi.org/10.5194/acp-13-2563-2013

Prinn, R., Huang, J., Weiss, R., Cunnold, D., Fraser, P., Simmonds, P., et al. (2005). Evidence for variability of atmospheric hydroxyl radicals over the past quarter century. Geophysical Research Letters, 32(7), L07809. http://doi.org/10.1029/2004GL022228

Rigby, M., Montzka, S. A., Prinn, R. G., White, J. W. C., Young, D., O'Doherty, S., et al. (2017). Role of atmospheric oxidation in recent methane growth. Proceedings of the National Academy of Sciences of the United States of America. http://doi.org/10.1073/pnas.1616426114

Rigby, M., Prinn, R. G., O'Doherty, S., Montzka, S. A., McCulloch, A., Harth, C. M., et al. (2013). Re-evaluation of the lifetimes of the major CFCs and CH3CCl3 using atmospheric trends. Atmospheric Chemistry and Physics, 13(5), 2691–2702. http://doi.org/10.5194/acp-13-2691-2013

Turner, A. J., Frankenberg, C., Wennberg, P. O., & Jacob, D. J. (2017). Ambiguity in

the causes for decadal trends in atmospheric methane and hydroxyl. Proceedings of the National Academy of Sciences of the United States of America, 44, 201616020. http://doi.org/10.1142/3171

---

## Short Comment (SC1) · 22 Jun 2018

As outlined in https://www.geoscientific-model-development.net/about/manuscript_types.html GMD is expecting that the model code is publicly available through a permanent arrangement. Given the impermanence of email addresses, GMD encourages authors acting as a point of contact for obtaining the code to improve the availability with a more permanent and public arrangement. When copyright or licensing restrictions prevent the public release of model code, or in the cases where there is some other good reason for not allowing public access to the code, authors need to state the reasons for why access is restricted and need to explain how access can be obtained (e.g. signing a license agree or join a consortium).

Lutz Gross GMD Executive Editor

---

## Referee Comment (RC2) · Anonymous Referee #2 · 25 Jun 2018

**Overview**

This paper describes advances to the NASA GISS ModelE2-Yale Interactive terrestrial Biosphere global chemistry-climate model (ModelE2-YIBs) and its use to optimise natural methane emission sources for the year 2005, through comparison of modelled and observed surface atmospheric methane concentrations. These emission inventories and the overall model performance are then assessed against atmospheric column methane measurements from the SCIAMACHY satellite instrument and ozone sonde measurements.

The Global Methane cycle continues to be a topic of much current interest. Methane is policy-relevant; it has the second largest radiative forcing after carbon dioxide and methane mitigation is an attractive option in achieving the warming targets of the Paris

[Figure]

Climate Agreement. The cited synthesis papers of Kirschke et al. [2013] and Saunois et al. [2016] both conclude that there is still significant uncertainty in the magnitude, temporal trends and spatial distributions of the different methane sources and sinks. The papers also highlight a significant gap between the total methane emissions derived from aggregating bottom-up, largely process-based estimates and the top-down estimates derived from atmospheric measurements.

With the uncertainty in the methane emission source terms, many chemistry-climate and Earth System models prescribe the surface atmospheric methane concentrations. The use of an interactive methane scheme (i.e. driven with surface methane emission and removal processes) is welcome, while technically challenging. Methane is relatively long-lived in terms of tropospheric chemistry, making it both sensitive to and affecting the hydroxyl radical concentrations. Thus successful modelling of methane needs a robust description of OH; a 1% change in OH concentration is equivalent to $\tilde{} 5$ Tg $CH_4$ yr$^{-1}$ change in methane emissions.

This is a limited study in that emission inventories and the model evaluation are for a single year (2005) and the derived inventories may be specific to the ModelE2-YIBs. It would have been more interesting to consider the inventories and model performance over a longer period (e.g., 2000-2014) covering both the period of near-zero growth between 2000 and 2006 and the renewed growth from 2007 onwards.

Although the paper falls in the remit of the journal, there are a number of key issues that need to be addressed before it can be considered for publication.

**Specific Comments**
Model development: The present model builds on the cited work of (1) Shindell et al. [2013], who described an interactive methane (and ozone) chemistry scheme in the GISS E2 chemistry-climate model, albeit for emissions from 2005 onwards, and (2) Yue and Unger [2015], who developed YIBS v1.0, a dynamic vegetation model for carbon-cycle studies, which also includes ozone-induced vegetation damage and

biogenic VOC emissions. The main model development appears to be coupling of the GISS E2 chemistry-climate and YIBS models.

The title of the paper gives the impression that significant advances have been in the representation of an interactive methane chemistry scheme in ModelE2-YIBs. In which case, I would have had many comments about comparing with other OH concentration datasets, using other atmospheric tracers ($CO$, $CO_2$) to constrain specific methane sources. In reality, it seems more limited. The only description of the model developments made in this paper (page 4, line 15) is in relation to the earlier work by Shindell et al. [2013] "this study updates the natural non-wetland methane fluxes; focuses on steady-state methane; applies a different anthropogenic emissions inventory; includes a new land surface model with interactive computation of isoprene and monoterpene emissions; and applies observed ocean boundary conditions". Although these are to some extent secondary to the primarily objective of the interactive methane scheme, there should nonetheless be some discussion as to how these have improved the overall model performance (compared to for example the GISS E2 chemistry-climate model), either in the main paper or as supplementary information. As an example, we are presented with global annual biogenic VOC emission estimates (Table 1, page 8), with no discussion as to how these compare to previous or other estimates (e.g., see Figure 10 in Sindelarova et al., 2014).

Wetland methane emissions: On page 10, line 19 and Table 2, a total of 140 Tg $CH_4$ $yr^{-1}$ is derived for the global mean emissions from wetlands for the year 2005. Effectively, this is a residual term after specifying all the other methane emission sources.

Earlier in the paper (page 4, line 5), the authors state "The model-measurement comparison was used to refine the spatial and temporal distribution of methane emissions from wetlands. The second and third steps were repeated, applying the newly optimized wetland emissions to ModelE2-YIBs, until strong model–measurement agreement was achieved". Later on the same page (line 12), "Using ModelE2, Shindell et al. [2013] previously used a similar procedure of modifying the wetland methane source to achieve a modeled methane concentration that is in line with present-day observations, noting that the accuracy of the magnitude of the wetland flux that is derived in this way depends on whether the other prescribed fluxes have been accurately assigned. (Relative to the Shindell et al. [2013] study, this study updates the natural non-wetland methane fluxes ....".

As far as I can tell, there is no further discussion of this optimisation process, what is involved, what is meant by strong agreement and hence how the the emission total of 140 Tg CH$_4$ yr$^{-1}$ is derived. The implication is that the wetland methane emissions are taken from or as used in Shindell et al. [2013]. This should be clarified and the text amended. I note that this optimised wetland emission dataset is provided in the Supplementary Information, as an annual dataset.

For sure, the total is within the range of current estimates (Saunois et al. [2016] is an update of and effectively supersedes Kirschke et al. [2013]). As someone who both derives and uses methane wetland emission datasets, the single annual dataset provided is of little value. We know that wetland methane emissions vary seasonally. I would like to see more information about the dataset, e.g., temporal trends (both seasonally and inter-annually), how do the regional totals compare to those in Table 4 of Saunois et al. [2016]? The wetland model intercomparison of Melton et al. [2013, cited paper] summarised the then state-of-the-art in wetland modelling and the large uncertainty in modelled wetland area and wetland methane emissions. To remove one of the largest areas of uncertainty (in wetland area), the wetland models contributing to the synthesis paper of Saunois et al. [2016] all used the same prescribed spatially and time-varying wetland product (SWAMPS), described in the follow-on paper of Poulter et al. [2017]. How do the wetland areas compare with SWAMPS?

Soil uptake: Similar comments can be made about the lack of information on

the soil uptake of methane. The spatial and temporal distributions given in Fung et al. [1991] are used (Page 7, line 25) and a total uptake of 60 Tg CH$_4$ yr$^{-1}$ is then derived (Table 2, page 9), which appears to influence the derived wetland methane emission estimates (page 10, line 19). This is then compared with and found to be higher than recent estimates (Page 10, lines 27-29). Could not the biome-specific measurements in the cited paper by Dutaur and Verchot not be used to create a new global methane uptake driven with relevant parameters from the land-surface model? More information is needed on how the total was derived.

Emission Maps: It would have been useful to include maps of the various methane emission sources (and any seasonal cycles) either in the paper or in the Supplementary Information to help interpret Figures 1, 3 and 4.

Model performance against observations: As presented, the model performance appears impressive, with differences of ˜ 1-2% between the modelled and observed methane concentrations (both surface and column). The OH field is considered to be realistic as it gives atmospheric methane lifetimes in agreement with other estimates. That said, there are issues.

The seasonal cycle at surface high latitude southern hemisphere sites is underestimated (Figure 3, pages 14-16). Is this because of the temporal and spatial assumptions made in the natural methane sources? (as well as the cited underestimation of the austral summertime chemical loss). The model fails to capture the annual cycle at a few locations, notably Pallas-Sammaltunturi in Finland; Barrow in Alaska, USA; and Ulaan Uul in Mongolia. This is ascribed to local influences. From Hayman et al. [2014], a similar study using the UK HadGEM2 chemistry-climate model with an interactive methane scheme, it is likely that the Barrow and Pallas-Sammaltunturi sites are (over)influenced by wetland emissions and the Ulaan Uul by other sources.

The model performance is slightly worse against the mean SCIAMACHY atmospheric

methane column mixing ratios (XCH4) (page 16, section 4.2). In their comparison, Hayman et al. (2014) also found that the HadGEM2 chemistry-climate model underestimated the observed SCIAMACHY XCH4, because the modelled $CH_4$ concentration fell off too rapidly with altitude (note the model configuration used a tropospheric chemistry scheme with an additional first-order loss process for methane to represent stratospheric methane chemistry, unlike the case here). It might not be a chemical problem (of sources and sinks) but potentially atmospheric dynamics and transport. This could be tested using other satellite $CH_4$ products which are more sensitive to the upper tropospher and lower stratosphere (e.g., TES, IASI).

VOC speciation (page 5, line 8): The chemical scheme has explicit representations of methane, isoprene, and formaldehyde, "while other hydrocarbons are represented using a lumped scheme (Houweling et al., 1998) that is based on the Carbon Bond Mechanism-4 (Gery et al., 1989) and the Regional Atmospheric Chemistry Model (Stockwell et al., 1997)". No information is given on how the non-methane VOC emissions are attributed to the model VOCs. Presumably, the existing NMVOC speciation is used.

Initialisation and Runtime: Nothing is said about how the model runs were initialised nor about runtime and platforms.

**Code availability**
The source code for ModelE2-YIBs (version 1.1) is available on request to the authors.

Included in the Supplementary Information as a zipped file are the natural methane emission and methane soil sink datasets, as gridded annual averages. I see little value in these as currently provided as several of the sources (e.g., wetlands) have strong seasonal variations. The EXCEL spreadsheet format should be converted into a non-proprietary format, e.g., flat text (such as comma separated variable) or more

usefully netCDF or similar.

**Technical comments**
Page 2, Line 27: "principle sink" should be "principal sink"

**References**
Fung et al., 1991: Three-dimensional model synthesis of the global methane cycle, J. Geophys. Res., 96, 13033-13065, doi:10.1029/91JD01247.

Hayman et al., 2014: Comparison of the HadGEM2 climate-chemistry model against in situ and SCIAMACHY atmospheric methane data, Atmos. Chem. Phys., 14, 13257-13280, doi:10.5194/acp-14-13257-2014.

Poulter et al., 2017: Global wetland contribution to 2000-2012 atmospheric methane growth rate dynamics, Env. Res. Lett.,12, 094013, doi: 10.1088/1748-9326/aa8391.

Saunois et al., 2016: The global methane budget 2000-2012, Earth Syst. Sci. Data, 8, 697-751, doi:10.5194/essd-8-697-2016.

Shindell et al., 2013: Interactive ozone and methane chemistry in GISS-E2 historical and future climate simulations. Atmos. Chem. Phys., 13, 2653-2689, doi:10.5194/acp-13-2653-2013

Sindelarova et al., 2014: Global data set of biogenic VOC emissions calculated by the MEGAN model over the last 30 years. Atmos. Chem. Phys., 14, 9317-9341, doi:10.5194/acp-14-9317-2014.

Yue and Unger, 2015: The Yale Interactive terrestrial Biosphere model version 1.0: description, evaluation and implementation into NASA GISS ModelE2. Geosci.

Model Dev., 8, 2399–2417, doi:10.5194/gmd-8-2399-2015.

---

## Author Comment (AC1) · 10 Sep 2018

**Response to Reviewer #1**

We thank the reviewer for their helpful comments, which have led us to an improved version of the paper. Here, the reviewer's comments are shown in boldfaced black text, and our responses are shown in non-boldfaced blue text. The page and line numbers to which we refer in our responses correspond to the updated manuscript (the comments of both reviewers are taken into account in this updated manuscript).

**Harper and coauthors present a global atmospheric chemistry-climate model with methane emissions. The paper documents the emissions of methane and other compounds, then evaluates the simulated concentrations of methane and ozone against observations. The methods are reasonable and the comparison to observations is sufficient to show that the model appears to be performing competently. The paper is written clearly. Some methods need greater explanation and discussion, which can be accomplished with modest revisions.**

1. **The model construction and budget analysis are based on the assumption that atmospheric methane was in steady state in 2005. Although the atmospheric methane concentrations were approximately stable during 2000-2007, as the authors say on p7, atmospheric methane may not have been in steady state at that time because emissions and OH may have been changing (Rigby et al. 2017; Turner et al., 2017). The steady state assumption, its limitations, and implications for model interpretation should be discussed.**

We agree with the reviewer that although the atmospheric methane concentrations were approximately stable during 2000–2007, atmospheric methane may not have been in steady state at that time because emissions and OH may have been changing at the same time.

We modified the description of the experimental set-up (Page 7, Line 22): "The E2005 simulation was run until atmospheric methane reached steady state, such that the global chemical sink came into balance with the net global source (prescribed sources minus prescribed soil sink), resulting in a relatively stable atmospheric methane abundance. Steady-state conditions were diagnosed using the global annual-mean atmospheric burden of methane. The final 10 years of the 45 year simulation are used for analysis. Year-to-year variation in the methane burden for the final 10 model years is < 3.2 Tg $CH_4$. Year-to-year variation in the global-average surface methane concentration is < 1.3 ppbv. The year of interest for this study, 2005, fell within a roughly 8 year period that witnessed a largely stable global-mean concentration of methane in Earth's atmosphere (Dlugokencky et al., 2009). The observed stability in the concentration of methane does not necessarily indicate temporally invariant global sources and sinks over this era (Rigby et al., 2017; Turner et al., 2017). For example, a recent analysis by Turner et al. (2017) suggests that simultaneous counterbalancing changes in methane emissions and loss to OH may be responsible for the observed stability in the methane concentration in the early 2000s. Therefore, the methane budget derived in this study by assuming steady state conditions represents just one plausible solution that can lead to a stable atmospheric methane concentration. This assumption is convenient in global chemistry–climate modeling where the simulated climate state does not correspond to an exact meteorological year. The derived solution is constrained by both the prescribed methane fluxes and other forcing data that can affect atmospheric methane, such as: emissions of other short-lived compounds; the prescribed ocean conditions, which influence the physical climate state; and the concentrations of the non-methane long-lived greenhouse gases, which influence the radiation budget.

The non-wetland natural methane fluxes that are prescribed are based on published estimates (Sect. 3) and are representative of the 2000s contemporary era but are not necessarily specific to year 2005. Likewise, the prescribed sea ice distribution and sea surface temperatures are observation-based five year means centered on year 2005. The derived methane budget, therefore, represents a 2000s climatology and is approximately, but not precisely, representative of year 2005 conditions."

2. **In the abstract and conclusions, the emissions magnitude and especially its partitioning into natural sources are stated too confidently and simply. These estimates assume that the prescribed anthropogenic emission inventory and the simulated CH4 loss are correct. Any error in these other budget terms would alter the authors' estimate of natural emissions. The emissions values should be presented as a best fit within the context of the other model assumptions.**

Following the reviewer's suggestion, in the Abstract (Page 1, Line 18), we have added this sentence: "The wetland methane flux is calculated as a best fit; thus, the accuracy of this derived flux assumes accurate simulation of methane chemical loss in the atmosphere and accurate prescription of the other methane fluxes (anthropogenic and natural)."

We have altered the first sentence of the conclusions section (Page 26, Line 11): "The results of the optimization process using atmospheric modeling indicate global annual methane emissions of 140 Tg $CH_4$ $y^{-1}$ from wetlands; **this derivation assumes accurate representation of the other methane fluxes and atmospheric chemical loss in the model.** The global annual methane emissions magnitude from all natural sources is 181 Tg $CH_4$ $y^{-1}$."

The manuscript does already describe the limitation in our ability to partition between the various methane sources. Following the reviewer's suggestion, we have further extended the key paragraph (Page 4, Line 16): "Using ModelE2, Shindell et al. (2013) previously used a similar procedure of modifying the wetland methane source to achieve a modeled methane concentration that is in line with present-day observations, noting that the accuracy of the magnitude of the wetland flux that is derived in this way depends on whether the other prescribed fluxes have been accurately assigned. **That is, the applied methodology calculates the wetland methane emission magnitude as a best fit under the assumption that the other methane fluxes and simulated atmospheric chemical loss are accurately represented in the global model.**" And here (Page 4, Line 21): "Relative to the Shindell et al. (2013) study, this study updates the natural non-wetland methane fluxes; applies a different anthropogenic emissions inventory; includes a new land surface model with interactive computation of isoprene and monoterpene emissions; and applies observed ocean boundary conditions. This methodology permits harmonization of the modeled methane mole fractions with contemporary observations, but can potentially misattribute the methane fluxes among the various source categories. Planned chemistry–climate simulations that will make use of the natural methane inventory developed here are specifically designed to investigate perturbations in anthropogenic methane emissions."

3. **The paper needs greater detail about how the natural methane emissions were optimized. The general approach is described a bit in Section 2, but lacks detail for a reader to attempt to reproduce it. I suggest providing this greater detail in Section 3. What observations were used in the optimization? Was it a formal optimization of some cost function or ad hoc trial and error with visual comparison? I would expect that the optimal emissions would produce an**

**unbiased global mean, but Section 4.1 reports and Figs 2 and 3 show that the model is systematically higher than observations at almost all sites.**

Please see Response to Reviewer #2 Point (3) for an updated description of the optimization methodology. We address the model–measurement comparisons in our response to the next point.

4. **In the abstract and elsewhere, 1% model bias against surface observations is acceptable, but not excellent. It may be comparable to the performance of other models, but it is one-fifth of the interhemispheric ratio NH/SH: 1.05. For a well-mixed gas like methane, a 1% model error after optimization is substantial.**

At no place in the manuscript do we refer to the model performance as "excellent." The reviewer may be confusing the different purposes of global chemistry–climate models (CCMs) versus chemistry–transport models (CTMs). We clarify this distinction here. We work with a global chemistry–climate model that has biases in the climate simulation itself (like all global climate models). Consider that we would actually be slightly more worried if we achieved an almost zero bias or an "excellent" comparison with observations. The ultimate purpose of a CCM is to study feedbacks and linkages between changes in atmospheric composition, radiation, and climate dynamics; there is a focus on understanding the role of interactive Earth system processes in determining the global climate sensitivity. In contrast, CTMs (with "correct meteorology," e.g., GEOS-Chem) can and must be used for formal optimization procedures to constrain surface emissions. We completely understand that for methane a 1% model error after a formal optimization process in a CTM would be considered substantial. That is not the case for a CCM. Certainly, we could go on and on applying additional iterations of our optimization process to further minimize discrepancies between modeled and measured methane mixing ratios. However, we argue that additional iterations are not justified at this point we have achieved (1) because this framework is for coupled CCM studies and (2) because of the existing limitations and uncertainties in model–measurement comparisons. Indeed, we show that our methane simulation is reasonable and realistic compared to and within the limitations of existing measurement comparisons.

5. **The supplement contains data in Excel xlsx format. I recommend an open source file format readable by free software, but I defer to the editor on whether this is required.**

We now use the comma-separated values (CSV) file format for all of the datasets included as part of the Supplementary Information.

**Minor comments**
6. **P2L19: CH4 is also oxidized by O(1D) in the stratosphere.**

We have added this methane sink to the indicated sentence. The Kirschke et al. (2013) reference, already cited in the original version of the sentence, covers this reaction, so no references were added (Page 2, Line 19): "Additional chemical loss occurs in the stratosphere via reactions with chlorine radicals and excited-state oxygen radicals ($O^1D$) (Kirschke et al., 2013; Portmann et al., 2012)."

**7. P5L13. Is version 1.1 a past model version or the new version described by this paper?**

Version 1.1 is the new version of the model that is described in this paper. We have improved the description of the various model versions to make this clear (Page 5, Line 22): "This paper describes the new version 1.1 of ModelE2-YIBs. ModelE2-YIBs version 1.1 refers to the use of interactive methane chemistry and dynamic methane emissions (including application of the final contemporary natural methane flux inventory described in Sect. 3) within the framework of ModelE2-YIBs version 1.0. ModelE2-YIBs version 1.0 refers to YIBs version 1.0 (Yue and Unger, 2015) coupled to the version of ModelE2 described by Schmidt et al. (2014)."

**8. P6L21. Were the LLGHG concentrations prescribed at the surface or also elsewhere?**

The concentrations are prescribed for the non-methane long-lived greenhouse gases (e.g., $CO_2$, $N_2O$, and CFCs) only in the first model layer (i.e., the layer closest to the surface). We have added the term "surface-level" to this sentence to clarify (Page 7, Line 3): "Prescribed global annual-mean surface-level mixing ratios of the non-methane well-mixed greenhouse gases are likewise from the RCP8.5 scenario (Meinshausen et al., 2011; Riahi et al., 2007): 379.3 ppmv $CO_2$, 319.4 ppbv $N_2O$, and 793 pptv chlorofluorocarbons (CFCs = CFC-11 + CFC-12)."

**9. P10L9 The work of Turner et al. is slightly misrepresented. The gross magnitude of methane emissions are well constrained, with uncertainty of 10% or less in the global total (Turner et al., 2017; also Prather et al., 2012). Turner et al. (2017) and also Rigby et al. (2017) showed that showed that observations poorly constrain partitioning and small but important trends in this total, although see Prather and Holmes (2017) for ways that exploiting spatial patterns could extract more information from existing observations.**

We have modified the sentence (Page 11, Line 7): "While the gross magnitude of methane emissions is well constrained, substantial uncertainties remain regarding the partitioning of methane emissions among source categories (Rigby et al., Turner et al., 2017). The interpretation of isotope composition measurements is currently ambiguous and complex (Turner et al., 2017). Prather and Holmes (2017) have recently suggested new approaches to extract more useful information from existing observations by exploiting spatial patterns."

We already have stated (Page 2, Line 32): "Together, these estimates provide a constraint on the total methane flux into the atmosphere; however, apportionment of this total into contributions from the individual source sectors is highly uncertain (Kirschke et al., 2013; Saunois et al., 2016)."

**10. P17L25. I believe the CH4 lifetime estimate by Rigby et al. (2013) should supersede Prinn et al. (2005), although the values are similar.**

The methane lifetimes against OH are similar from the two references: 10.6 ± 0.4 years from Rigby et al. (2013) and 10.2 (+0.9, -0.7) years from Prinn et al. (2005). The estimate by Rigby et al. (2013) is based upon the same general modeling framework as is used by Prinn et al. (2005), and therefore can be considered to be an update of the earlier work. We now use the methane lifetime estimate made by Rigby et al. (2013) in place of that made by Prinn et al. (2005):

In Sect. 4.3, we use the Rigby et al. (2013) estimates in place of the Prinn et al. (2005) estimates (Page 22, Line 29): "
[revised manuscript text omitted]

---

## Author Comment (AC2) · 10 Sep 2018

**Response to Lutz Gross, GMD Executive Editor**

**As outlined in https://www.geoscientific-model-development.net/about/manuscript_ types.html GMD is expecting that the model code is publicly available through a permanent arrangement. Given the impermanence of email addresses, GMD encourages authors acting as a point of contact for obtaining the code to improve the availability with a more permanent and public arrangement. When copyright or licensing restrictions prevent the public release of model code, or in the cases where there is some other good reason for not allowing public access to the code, authors need to state the reasons for why access is restricted and need to explain how access can be obtained (e.g. signing a license agree or join a consortium).**

We have updated the "Code and data availability" section (Page 27, Line 4): "The source code for the site-level YIBs model version 1.0 is available at https://github.com/YIBS01/YIBS_site. The GISS ModelE2 source code can be obtained from NASA GISS (https://www.giss.nasa.gov/tools/modelE/). Included as supplemental information are the gridded natural methane fluxes and the numerical model output used to make the figures. Gridded files of natural methane fluxes associated with the Fung et al. (1991) dataset were obtained from NASA GISS (data.giss.nasa.gov/ch4_fung). Column-averaged methane concentrations from SCIAMACHY were obtained from the University of Bremen (iup.uni-bremen.de/sciamachy/NIR_NADIR_WFM_DOAS/products/). Other data used as model input or for analysis of model output are listed in the references."

Reference:

Fung, I., John, J., Lerner, J., Matthews, E., Prather, M., Steele, L.P., and Fraser, P.J.: Three-dimensional model synthesis of the global methane cycle, J. Geophys. Res., 96, 13,033-13,065, doi: 10.1029/91JD01247, 1991.

---

## Author Comment (AC3) · 10 Sep 2018

The comment was uploaded in the form of a supplement: https://www.geosci-model-dev-discuss.net/gmd-2018-85/gmd-2018-85-AC3-supplement.pdf

---

## Author Comment (AC4) · 10 Sep 2018

**Response to Reviewer #2**

We thank the reviewer for their thoughtful comments. Here, the reviewer's comments are shown in boldfaced black text, and our responses are shown in non-boldfaced blue text. The page and line numbers to which we refer in our responses correspond to the updated manuscript (the comments of both reviewers are taken into account in this updated manuscript).

**Overview**

**This paper describes advances to the NASA GISS ModelE2-Yale Interactive terrestrial Biosphere global chemistry-climate model (ModelE2-YIBs) and its use to optimise natural methane emission sources for the year 2005, through comparison of modelled and observed surface atmospheric methane concentrations. These emission inventories and the overall model performance are then assessed against atmospheric column methane measurements from the SCIAMACHY satellite instrument and ozone sonde measurements.**

**The Global Methane cycle continues to be a topic of much current interest. Methane is policy-relevant; it has the second largest radiative forcing after carbon dioxide and methane mitigation is an attractive option in achieving the warming targets of the Paris Climate Agreement. The cited synthesis papers of Kirschke et al. [2013] and Saunois et al. [2016] both conclude that there is still significant uncertainty in the magnitude, temporal trends and spatial distributions of the different methane sources and sinks. The papers also highlight a significant gap between the total methane emissions derived from aggregating bottom-up, largely process-based estimates and the top-down estimates derived from atmospheric measurements.**

**With the uncertainty in the methane emission source terms, many chemistry-climate and Earth System models prescribe the surface atmospheric methane concentrations. The use of an interactive methane scheme (i.e. driven with surface methane emission and removal processes) is welcome, while technically challenging. Methane is relatively long-lived in terms of tropospheric chemistry, making it both sensitive to and affecting the hydroxyl radical concentrations. Thus successful modelling of methane needs a robust description of OH; a 1% change in OH concentration is equivalent to~ 5 Tg CH4 yr$^{-1}$ change in methane emissions.**

We fully agree with the reviewer's comments.

1. **This is a limited study in that emission inventories and the model evaluation are for a single year (2005) and the derived inventories may be specific to the ModelE2-YIBs. It would have been more interesting to consider the inventories and model performance over a longer period (e.g., 2000-2014) covering both the period of near-zero growth between 2000 and 2006 and the renewed growth from 2007 onwards.**

Please see response to Reviewer #1 Point (4). Our approach applies a CCM as a tool for investigating Earth system processes and global climate sensitivity, not a CTM approach (with "correct" reanalysis meteorology) for constraining emission sources. We agree that the derived wetland methane inventory is somewhat specific to ModelE2-YIBs; the other methane sources are derived from published estimates, as described in Sect. 3. A major technical advance is coupling the dynamic methane simulation with the YIBs terrestrial biosphere. This capability allows us to investigate, for instance, the impacts of changes in the terrestrial biosphere on methane and the impacts of anthropogenic methane mitigation on the

terrestrial biosphere, within the context of global climate change. We will apply the framework to examine the full Earth system impacts of mitigating anthropogenic methane emissions. The overall methane inventory is highly relevant to other global models. Interactive methane simulations are computationally expensive and time consuming. When our cluster was operating at peak performance, each 45-year simulation required around 2–3 weeks of run time; actual time from initial submission to simulation completion, accounting for resubmissions of the job and time spent in the simulation queue, was typically several weeks longer than the base run time. Our dataset, optimized for our model, can serve as a useful initial dataset for other models. Starting with a close approximation of prescribed methane fluxes can reduce the computational power and time needed for optimization, perhaps prompting more widespread use of interactive methane schemes in global modeling.

We also agree that the study suggested by the reviewer (examining a longer decadal period 2000–2014, extending to the period of growth in atmospheric methane that occurred after 2007) is an interesting one to consider that is most appropriately tackled using a global CTM approach. In our study, we run time-slice simulations, in which the applied forcing datasets (including the prescribed methane fluxes) are repeated for each model year of the run. We run the simulation long enough to allow methane to achieve a stable concentration. Page 7, Line 25: "Year-to-year variation in the methane burden for the final 10 model years is < 3.2 Tg $CH_4$. Year-to-year variation in the global-average surface methane concentration is < 1.3 ppbv." Our year of interest, 2005, falls within a period when the atmospheric methane concentration was largely stable. We are currently using the optimized methane scheme from this study in time-slice simulations aimed at probing the impacts of anthropogenic methane emission perturbations on atmospheric concentrations of a suite of short-lived climate pollutants. The optimized methane scheme developed in our current study serves as a useful starting point both (1) for optimizing the methane schemes in other models and (2) for setting up transient simulations, in which the prescribed methane emissions evolve over time.

We have made the following updates to the conclusions section (Sect. 6):

(1) We have added the words "time-slice" to the following sentence (Page 26, Line 23): "The improved methane scheme is currently being applied to time-slice chemistry–climate simulations to quantify the methane response and concomitant radiative forcing associated with perturbations in anthropogenic methane emissions."

(2) We have added to the end of Sect. 6 (Page 26, Line 25): "The gridded natural methane fluxes associated with the optimized methane scheme in ModelE2-YIBs are provided in the Supplemental Information. This dataset can serve as a useful starting point for optimization of the interactive methane schemes in other atmospheric models. Starting with a reasonable approximation of prescribed methane fluxes can reduce the computational power and time needed for optimization in other models, potentially prompting more widespread use of interactive methane schemes in global modeling. The optimized methane inventory developed in this study additionally serves as a useful starting point for a potential follow-up study aimed at optimization for transient simulations, in which the prescribed methane emissions evolve over time."

Please also see our response to point (1) of Reviewer #1; we have added a description to the paper describing that our derived methane budget represents a 2000s climatology, centered around year 2005.

Although the paper falls in the remit of the journal, there are a number of key issues that need to be addressed before it can be considered for publication.

**Specific Comments**

2. Model development: The present model builds on the cited work of (1) Shindell et al. [2013], who described an interactive methane (and ozone) chemistry scheme in the GISS E2 chemistry-climate model, albeit for emissions from 2005 onwards, and (2) Yue and Unger [2015], who developed YIBS v1.0, a dynamic vegetation model for carbon-cycle studies, which also includes ozone-induced vegetation damage and biogenic VOC emissions. The main model development appears to be coupling of the GISS E2 chemistry-climate and YIBS models.

The title of the paper gives the impression that significant advances have been in the representation of an interactive methane chemistry scheme in ModelE2-YIBs. In which case, I would have had many comments about comparing with other OH concentration datasets, using other atmospheric tracers (CO, CO2) to constrain specific methane sources. In reality, it seems more limited. The only description of the model developments made in this paper (page 4, line 15) is in relation to the earlier work by Shindell et al. [2013] "this study updates the natural non-wetland methane fluxes; focuses on steady-state methane; applies a different anthropogenic emissions inventory; includes a new land surface model with interactive computation of isoprene and monoterpene emissions; and applies observed ocean boundary conditions". Although these are to some extent secondary to the primarily objective of the interactive methane scheme, there should nonetheless be some discussion as to how these have improved the overall model performance (compared to for example the GISS E2 chemistry-climate model), either in the main paper or as supplementary information. As an example, we are presented with global annual biogenic VOC emission estimates (Table 1, page 8), with no discussion as to how these compare to previous or other estimates (e.g., see Figure 10 in Sindelarova et al., 2014).

The reviewer is correct that a major technical advance is coupling the dynamic methane simulation to the YIBs terrestrial biosphere. Shindell et al. (2013) and the GISS ModelE2 AR5 version use a default GISS land cover dataset and vegetation representation (e.g., Matthews, Global vegetation and land use: New high-resolution data bases for climate studies, J. Clim. Appl. Meteorol., 1983). Yue and Unger (2015) describe the coupling of the YIBs model to the GISS ModelE2 chemistry–climate model (Yue and Unger: The Yale Interactive terrestrial Biosphere model version 1.0: description, evaluation and implementation into NASA GISS ModelE2, Geosci. Model Dev., 2015). The simulations presented in Yue and Unger (2015) were run using prescribed methane concentrations, as opposed to running with interactive methane chemistry and dynamic methane emissions, such as we apply here. The coupling between ModelE2-YIBs and the dynamic methane simulation is not trivial. The new framework will allow us to examine how changes in anthropogenic methane emissions impact the terrestrial biosphere and, in turn, how changes in the terrestrial biosphere impact atmospheric methane.

We have clarified how ModelE2-YIBs version 1.1 relates to YIBs and ModelE2 (Page 5, Line 22; also see point (7) from Reviewer #1): "This paper describes the new version 1.1 of ModelE2-YIBs. ModelE2-YIBs version 1.1 refers to the use of interactive methane chemistry and dynamic methane emissions (including application of the final contemporary natural methane flux inventory described in Sect. 3) within the framework of ModelE2-YIBs version 1.0. ModelE2-YIBs version 1.0 refers to YIBs version 1.0 (Yue and Unger, 2015) coupled to the version of ModelE2 described by Schmidt et al. (2014)."

We have added the Yue and Unger (2015) and Unger et al. (2013) references to this sentence (Page 4, Line 21): "Relative to the Shindell et al. (2013) study, this study updates the natural non-wetland methane fluxes; applies a different anthropogenic emissions inventory; includes a new land surface model with interactive computation of isoprene and monoterpene emissions (Unger et al., 2013; Yue and Unger, 2015); and applies observed ocean boundary conditions."

3. Wetland methane emissions: On page 10, line 19 and Table 2, a total of 140 Tg CH4 yr$^{-1}$ is derived for the global mean emissions from wetlands for the year 2005. Effectively, this is a residual term after specifying all the other methane emission sources.

Earlier in the paper (page 4, line 5), the authors state "The model-measurement comparison was used to refine the spatial and temporal distribution of methane emissions from wetlands. The second and third steps were repeated, applying the newly optimized wetland emissions to ModelE2-YIBs, until strong model–measurement agreement was achieved". Later on the same page (line 12), "Using ModelE2, Shindell et al. [2013] previously used a similar procedure of modifying the wetland methane source to achieve a modeled methane concentration that is in line with present-day observations, noting that the accuracy of the magnitude of the wetland flux that is derived in this way depends on whether the other prescribed fluxes have been accurately assigned. (Relative to the Shindell et al. [2013] study, this study updates the natural non-wetland methane fluxes ....".

As far as I can tell, there is no further discussion of this optimisation process, what is involved, what is meant by strong agreement and hence how the the emission total of 140 Tg CH4 yr$^{-1}$ is derived. The implication is that the wetland methane emissions are taken from or as used in Shindell et al. [2013]. This should be clarified and the text amended. I note that this optimised wetland emission dataset is provided in the Supplementary Information, as an annual dataset.

For sure, the total is within the range of current estimates (Saunois et al. [2016] is an update of and effectively supersedes Kirschke et al. [2013]). As someone who both derives and uses methane wetland emission datasets, the single annual dataset provided is of little value. We know that wetland methane emissions vary seasonally. I would like to see more information about the dataset, e.g., temporal trends (both seasonally and inter-annually), how do the regional totals compare to those in Table 4 of Saunois et al. [2016]? The wetland model intercomparison of Melton et al. [2013, cited paper] summarised the then state-of-the-art in wetland modelling and the large uncertainty in modelled wetland area and wetland methane emissions. To remove one of the largest areas of uncertainty (in wetland area), the wetland models contributing to the synthesis paper of Saunois et al. [2016] all used the same prescribed spatially and time-varying wetland product (SWAMPS), described in the follow-on paper of Poulter et al. [2017]. How do the wetland areas compare with SWAMPS?

We have expanded both our methodological description and evaluation of the methane fluxes:

(1) In Sect. 2, when we introduce the optimization approach for wetlands, we have added text to point the reader to Sect. 3 for additional information (Page 4, Line 10): "The resulting natural methane emissions inventory is described in Sect. 3, **along with additional details about the optimization process for the wetland methane source**."

(2) We do not use the wetland methane emissions from the cited Shindell et al. (2013) paper. In Sect. 3, we have added a detailed description of the optimization process used to derive the wetland methane emissions (Page 11, Line 28): "The iterative refinement process used to optimize the wetland methane flux was largely a trial-and-error based methodology that made use of literature-derived estimates and surface observations. The wetland methane flux is calculated as a best fit following prescription of the other fluxes. The baseline wetland methane emissions applied to the optimization process are the methane emissions from bogs and swamps from Fung et al. (1991); the magnitude, spatial distribution, and temporal distribution of these emissions were subsequently modified to varying degrees during the optimization process. At each step of the process, the annual cycle of modeled surface-level methane concentration was compared to observations from the NOAA ESRL measurement network at 50 globally distributed sites (Dlugokencky et al., 2015). The aim of the optimization process was to minimize the absolute value of the normalized mean bias (NMB) at the largest number of sites. Considering the full set of 50 sites, the final optimized scenario results in NMBs ranging from -1.3% (model underestimate) to +3.0% (model overestimate), with a median of +0.4%. At three quarters of sites, the NMB is between -1% and +1%. An evaluation of the simulated atmospheric methane distribution associated with the final optimized emissions inventory, including a comparison to SCIAMACHY methane columns (Schneising et al., 2009), is provided in Sect. 4. Modification of the temporal distribution of wetland methane emissions was guided by both the annual cycles of surface methane concentrations at the 50 NOAA ESRL measurement sites (Dlugokencky et al., 2015) and the annual cycle of wetland methane emissions simulated by the models participating in the WETCHIMP analysis (Melton et al., 2013).

The best fit of modeled atmospheric methane relative to the NOAA ESRL surface methane observations corresponds to the following modification of the baseline wetland methane emissions dataset. First, the baseline wetland methane emissions (extratropical bogs and tropical swamps) from Fung et al. (1991) were scaled to achieve an extratropical emissions fraction of 30% and a prescribed global emission magnitude of about 130 Tg $CH_4$ $y^{-1}$. A single scaling factor was applied in each grid cell in each month to the emissions from bogs; likewise, a separate single scaling factor was applied in each grid cell in each month to the emissions from swamps. For the WETCHIMP study, the mean extratropical emissions fraction among all participating models is about 30% (Melton et al., 2013). Secondly, an additional 10 Tg $CH_4$ $y^{-1}$ was added to the wetland methane emissions: (1) 2 Tg $CH_4$ $y^{-1}$ was added to 20°N–40°N over the months March through September; (2) 2 Tg $CH_4$ $y^{-1}$ was added to 0°–20°N over the months May through October; and (3) 6 Tg $CH_4$ $y^{-1}$ was added to 20°S–0° over all months. Finally, the seasonal cycle of the wetland methane emission hotspots in Finland and Russia (50°N–70°N) were adjusted: 0.5 Tg $month^{-1}$ decrease for each of June, July, and August; 0.65 Tg $month^{-1}$ increase in both September and October, and 0.2 Tg $month^{-1}$ increase in November."

(3) At the end of Sect. 3, we have added (1) new Figure 1 (Page 13 and shown below) and (2) information on the seasonal variation of wetland methane emissions (Page 12, Line 32): "The annual cycle of wetland methane emissions is plotted in Fig. 1. Monthly emissions are shown for the same latitudinal zones that are plotted in Melton et al. (2013) for six models participating in the WETCHIMP analysis (their Fig. 6, corresponding to the mean annual cycle for years 1993–2004). Global monthly methane emissions from wetlands range from 7.4–18.2 Tg $month^{-1}$ (Fig. 1). Monthly emissions show little variability from November to April (range: 7.4–9.5 Tg $month^{-1}$), followed by increasing emissions starting in May (12.9 Tg $month^{-1}$). Peak monthly emissions occur in July (18.2 Tg $month^{-1}$). The six WETCHIMP models simulate peak emissions, variously occurring between June and August, of slightly higher magnitude (approximate range for the six models: 20–35 Tg $month^{-1}$; Melton et al., 2013). The annual cycle of emissions for the 40°N–90°N latitudinal band is similar in shape to that for global emissions, with peak monthly emissions likewise occurring in July (9.1 Tg $month^{-1}$; Fig. 1). Monthly

emissions for the 20°N–40°N band show little variation throughout the year and are of low magnitude (range: 0.5–0.9 Tg month$^{-1}$; Fig. 1), while the WETCHIMP models generally exhibit a small peak on the order of 5 Tg month$^{-1}$ in this band in the Northern Hemisphere summer (Melton et al., 2013). The 0°–20°N band shows increasing monthly emissions between February and August, followed by declining monthly emissions (Fig. 1). The 20°S–0° band shows the largely opposite cycle, with minimum monthly emissions occurring in August (1.4 Tg month$^{-1}$). Monthly emissions from the tropics, considering 30°S–30°N, are largely consistent throughout the year, ranging from 6.0–8.0 Tg month$^{-1}$."

[Figure]

**Figure 1:** Monthly wetland methane emissions (Tg CH$_4$ month$^{-1}$) for several latitudinal bands for the optimized inventory.

(4) We have also added new Fig. 2 (Page 14 and shown below), new Fig. S1, and more information on the zonal distribution of methane fluxes (Page 14, Line 1): "The zonal distribution of annual wetland methane emissions is shown in Fig. 2, with emissions aggregated over 2°-latitude bands. Peak annual emissions occur near the equator, similar to the WETCHIMP multi-model mean (Melton et al., 2013, their Fig. 5, although shown in 3°-latitude bands). In the Southern Hemisphere, the optimized wetland methane inventory exhibits smaller secondary peaks near 15°S and 30°S. The WETCHIMP multi-model mean likewise exhibits regional peaks in these locations, but the magnitude of the peak at 30°S relative to the peak at the equator is stronger in the optimized inventory than in the WETCHIMP analysis. Like the WETCHIMP multi-model mean, the optimized wetland emissions inventory shows a wide secondary peak centered around 55°N. The secondary peak at 10°N is also seen in the WETCHIMP multi-model mean; in the optimized inventory, this peak exhibits a stronger magnitude relative to the main peak at the equator than occurs in the WETCHIMP analysis. The spatial distributions of the monthly wetland

methane emissions are shown in Fig. S1, and the gridded optimized monthly wetland methane emissions data are provided in the Supplementary Information."

[Figure]

**Figure 2:** Annual zonally summed wetland methane emissions (Tg CH$_4$ 2°-latitude band$^{-1}$ y$^{-1}$) for the optimized inventory.

(5) We have added new Table 3 (Page 15 and shown below), Fig. S2, and more information regarding the regional distribution of total methane emissions in the optimized inventory (Page 14, Line 18): "Total annual methane emissions from all non-oceanic sources are shown in Table 3 for 14 regions. Regional definitions follow Saunois et al. (2016). In their Table 4, Saunois et al. (2016) provide estimates of annual methane emissions (means for 2000–2009) for the same 14 regions, including both best estimates and ranges resulting from a set of inversions. The regional methane emissions from the optimized inventory fall within the suggested range of Saunois et al. (2016) for nine regions: temperate South America, tropical South America, central North America, boreal North America, southern Africa, northern Africa, Europe, China, and Oceania. For two other regions (contiguous USA and India), the emissions fall within 1–2 Tg y$^{-1}$ of the suggested range. Emissions in Southeast Asia from the optimized inventory are slightly lower than the range of 54–84 Tg y$^{-1}$ suggested by Saunois et al. (2016). The optimized inventory exhibits emissions that are higher than the suggested ranges of Saunois et al. (2016) for two regions: (1) Russia (suggested range: 32–44 Tg y$^{-1}$) and (2) Central Eurasia and Japan (suggested range: 38–51 Tg y$^{-1}$). For both regions, the strong emissions in the inventory applied here are associated with strong energy sector emissions and, in the case of Russia, strong wetland emissions. Comparison of simulated column-average methane concentrations with those from SCIAMACHY (Sect. 4.2) shows model underestimates on the order of 2% in these regions, which is typical of model underestimates in other regions. The global distributions of annual methane emissions by source category are shown in Fig. S2."

(6) We have added the bolded part to this statement (Page 15, Line 11): "The total emission magnitude of methane for 2005 in the ModelE2-YIBs inventory is 532 Tg y$^{-1}$ (Table 2), which corresponds well to the top-down estimate (548 Tg y$^{-1}$, range: 526–569 Tg y$^{-1}$) reported by the Kirschke et al. (2013) review **and**

**is only slightly outside of the range from the top-down estimate (552 Tg y$^{-1}$, range: 535–566 Tg y$^{-1}$) reported by the more recent Saunois et al. (2016) review.**"

**Table 3:** Regional annual methane emissions from non-oceanic sources (Tg y$^{-1}$). Regional definitions follow Saunois et al. (2016).

| Region | Annual methane emissions (Tg y$^{-1}$) |
|---|---|
| Temperate South America | 23.0 |
| Tropical South America | 70.4 |
| Central North America | 12.1 |
| Contiguous USA | 37.0 |
| Boreal North America | 17.7 |
| Southern Africa | 37.8 |
| Northern Africa | 38.4 |
| Europe | 30.6 |
| Russia | 60.7 |
| Central Eurasia and Japan | 57.2 |
| China | 50.5 |
| India | 26.3 |
| Southeast Asia | 47.4 |
| Oceania | 17.1 |

In the configuration applied in this study, the model assigns methane fluxes from wetlands using gridded input files. Page 7, Line 17: "For most sectors, anthropogenic and natural methane emissions are prescribed in the climate model using global, gridded input files; lake, oceanic, and terrestrial geological methane emissions are internally calculated by the model through prescription of emission factors in the model source code." That is, the model does not use an interactive wetland scheme, where wetland extent is calculated internally based on climate or other conditions. Neither does the model make use of any type of prescribed wetland extent product. Therefore, we are not able to make any type of comparison with the SWAMPS product (Poulter et al., 2017) mentioned by the reviewer.

In this study, we develop the natural methane flux dataset so that we can apply it to interactive methane model studies aimed at investigating the impacts of anthropogenic methane emission perturbations on atmospheric composition and radiative forcing. Page 4, Line 25: "Planned chemistry–climate simulations that will make use of the natural methane inventory developed here are specifically designed to investigate perturbations in anthropogenic methane emissions." In such studies, we are interested in the annual-mean impacts. For these studies, we apply the same natural methane fluxes to each model year because we want to isolate the impacts arising only from anthropogenic emissions perturbations. (We do apply monthly varying wetland methane emissions, but the same annual cycle is applied for each model year.) Therefore, we do not apply interannual variation in the wetland methane emissions; rather, we use the consistent 2000s climatology of natural methane fluxes (roughly year 2005) developed here for each model year in those simulations.

4. **Soil uptake: Similar comments can be made about the lack of information on the soil uptake of methane. The spatial and temporal distributions given in Fung et al. [1991] are used (Page 7, line 25) and a total uptake of 60 Tg CH4 yr$^{-1}$ is then derived (Table 2, page 9), which appears to influence the derived wetland methane emission estimates (page 10, line 19). This is then compared with and found to be higher than recent estimates (Page 10, lines 27-29). Could not the biome-specific measurements in the cited paper by Dutaur and Verchot not be used to create a new global methane uptake driven with relevant parameters from the land-surface model? More information is needed on how the total was derived.**

The reviewer raises an excellent idea for a future PhD project to develop and interrogate the methane soil sink with the few available measurements. In this current work with the global CCM, we assume that the anthropogenic emissions inventory is correct as a starting point.

As described on Page 12, Line 23: "The methane soil sink in the ModelE2-YIBs inventory corresponds to the top end of the range suggested by Dutaur and Verchot (2007) …" Thus, the soil sink is based on literature values. To be scientifically balanced, we then give additional context, suggesting that other reviews suggest lower methane uptake by soils: "… and is higher than the magnitude reported in recent reviews (e.g., top-down range: 26–42 Tg y-1; bottom-up range: 9–47 Tg y-1; Kirschke et al., 2013). However, the simulated total atmospheric lifetime of methane and the simulated methane mixing ratio in ModelE2-YIBs are well aligned with observation-based estimates (Sect. 4), suggesting that the overall rate of removal of methane is well represented in the model." We change "and is higher than" to "but is higher than" to underscore this point.

We have added another two sentences to reiterate that the derived wetland methane emissions depend on the prescribed soil uptake (and also depends on the magnitudes of the other prescribed methane emission sectors, Page 12, Line 25): "The wetland methane emissions are derived as a best fit given the other prescribed emissions, the methane soil sink, and the simulated chemical sink. Applying a weaker soil sink would have resulted in a lower magnitude for the derived wetland methane emissions; applying a stronger soil sink would have resulted in a higher magnitude for the derived wetland methane emissions."

Please also see response to point (2) from Reviewer #1.

5. **Emission Maps: It would have been useful to include maps of the various methane emission sources (and any seasonal cycles) either in the paper or in the Supplementary Information to help interpret Figures 1, 3 and 4.**

We have added such plots to the Supplementary Information, as described in our extended analysis in response to point (3) above.

6. **Model performance against observations: As presented, the model performance appears impressive, with differences of ˜ 1-2% between the modelled and observed methane concentrations (both surface and column). The OH field is considered to be realistic as it gives atmospheric methane lifetimes in agreement with other estimates. That said, there are issues.**

The seasonal cycle at surface high latitude southern hemisphere sites is underestimated (Figure 3, pages 14-16). Is this because of the temporal and spatial assumptions made in the natural methane sources? (as well as the cited underestimation of the austral summertime chemical loss). The model fails to capture the annual cycle at a few locations, notably Pallas-Sammaltunturi in Finland; Barrow in Alaska, USA; and Ulaan Uul in Mongolia. This is ascribed to local influences. From Hayman et al. [2014], a similar study using the UK HadGEM2 chemistry-climate model with an interactive methane scheme, it is likely that the Barrow and Pallas-Sammaltunturi sites are (over)influenced by wetland emissions and the Ulaan Uul by other sources.

The model performance is slightly worse against the mean SCIAMACHY atmospheric methane column mixing ratios (XCH4) (page 16, section 4.2). In their comparison, Hayman et al. (2014) also found that the HadGEM2 chemistry-climate model underestimated the observed SCIAMACHY XCH4, because the modelled CH4 concentration fell off too rapidly with altitude (note the model configuration used a tropospheric chemistry scheme with an additional first-order loss process for methane to represent stratospheric methane chemistry, unlike the case here). It might not be a chemical problem (of sources and sinks) but potentially atmospheric dynamics and transport. This could be tested using other satellite CH4 products which are more sensitive to the upper tropospher and lower stratosphere (e.g., TES, IASI).

We have added:

(1) Page 18, Line 4: "The model–measurement differences in annual cycles might also be associated with the temporal and spatial assumptions made in the prescribed methane emissions inventory."

(2) Page 18, Line 9: "Based on interactive methane simulations with the HadGEM2 chemistry–climate model, Hayman et al. (2014) likewise found model–measurement discrepancies in the annual cycles at these (and other) sites, finding that, in their simulations, the Barrow and Pallas-Sammaltunturi sites are strongly influenced by emissions from wetlands, while the Ulaan Uul site is influenced by other non-wetland emission sources."

(3) Page 22, Line 14: "Using interactive methane simulations in the HadGEM2 chemistry–climate model, Hayman et al. (2014) likewise found that the model underestimated column-averaged methane concentrations relative to SCIAMACHY observations due to simulated methane concentrations that decreased too rapidly with increasing altitude. The HadGEM2 simulations applied an explicit methane loss term to represent stratospheric methane oxidation (Hayman et al., 2014), while ModelE2 uses fully coupled dynamic stratospheric chemistry (e.g., Shindell et al., 2006)."

(4) Page 22, Line 20 (added bolded portion): "The model slightly overestimates annual-mean surface methane at 80 % of the NOAA ESRL measurement locations and underestimates column-averaged methane at most locations on the globe. This mis-match could indicate that the principal chemical sink of methane – reaction with OH – is slightly too strong in the model outside of the surface layer, **or it could indicate potential issues with the transport mixing rate of methane in the free troposphere and stratosphere. Future work with other vertically resolved satellite data products may help unravel the chemical and/or dynamical causes.** Overall, the model shows good agreement with measured methane mixing ratios, providing confidence in its ability to simulate the principal chemical and dynamical processes that affect methane in the atmosphere."

7. **VOC speciation** **(page 5, line 8): The chemical scheme has explicit representations of methane, isoprene, and formaldehyde, "while other hydrocarbons are represented using a lumped scheme (Houweling et al., 1998) that is based on the Carbon Bond Mechanism-4 (Gery et al., 1989) and the Regional Atmospheric Chemistry Model (Stockwell et al., 1997)". No information is given on how the non-methane VOC emissions are attributed to the model VOCs. Presumably, the existing NMVOC speciation is used.**

We have added the bolded statements (Page 5, Line 12): "The troposphere includes standard $NO_X$-$O_X$-$HO_X$-CO-$CH_4$ chemistry; methane, isoprene, **monoterpenes (as $\alpha$-pinene)**, and formaldehyde are explicitly represented in the model, while other hydrocarbons are represented using a lumped scheme (Houweling et al., 1998) that is based on the Carbon Bond Mechanism-4 (Gery et al., 1989) and the Regional Atmospheric Chemistry Model (Stockwell et al., 1997). **More recent updates to the chemical mechanism are described by Shindell et al. (2006, 2013). The alkane and alkene lumped hydrocarbon classes used in the ModelE2-YIBs chemical mechanism are calculated from the total NMVOC emissions from the prescribed emissions scenario (described in Sect. 2.2) by applying spatially explicit alkane-to-total-NMVOC and alkene-to-total-NMVOC ratios from the RCP8.5 inventory (Riahi et al., 2011) for year 2005.**"

8. **Initialisation and Runtime****: Nothing is said about how the model runs were initialised nor about runtime and platforms.**

To describe the atmospheric methane distribution defined at initialization, we added (Page 7, Line 13): "For simulations using the interactive methane scheme in ModelE2, the atmospheric methane distribution at initialization is defined through application of a vertical gradient, derived from HALOE observations (e.g., Russell et al., 1993), to prescribed hemispheric-mean surface methane concentrations (Dlugokencky et al., 2015)."

And on Page 6, Line 12: "The simulations were performed on the Omega cluster at the Yale Center for Research Computing (https://research.computing.yale.edu/support/hpc/clusters/omega). Omega is a 704-node 5632-core cluster based on the Intel Nehalem nodes and equipped with 36GB of RAM per node, a QDR Infiniband interconnect, and a high-speed Lustre DDN file system for parallel I/O. When the cluster was operating at peak performance, NASA ModelE2-YIBs had a runtime of 8–10 model days per hour using 88 processors."

9. **Code availability**
**The source code for ModelE2-YIBs (version 1.1) is available on request to the authors.**

**Included in the Supplementary Information as a zipped file are the natural methane emission and methane soil sink datasets, as gridded annual averages. I see little value in these as currently provided as several of the sources (e.g., wetlands) have strong seasonal variations. The EXCEL spreadsheet**

**format should be converted into a non-proprietary format, e.g., flat text (such as comma separated variable) or more usefully netCDF or similar.**

We now use .csv files for all of the datasets included as part of the Supplementary Information, and we now provide the wetland methane emissions at monthly resolution.

**Technical comments**
    **10.  Page 2, Line 27: "principle sink" should be "principal sink"**

Fixed.

[revised manuscript text omitted]